# Cross-talk between PRMT1-mediated methylation and ubiquitylation on RBM15 controls RNA splicing

Li Zhang[1†], Ngoc-Tung Tran[1†], Hairui Su[1†], Rui Wang[2], Yuheng Lu[3], Haiping Tang[4], Sayura Aoyagi[5], Ailan Guo[5], Alireza Khodadadi-Jamayran[1], Dewang Zhou[1], Kun Qian[6], Todd Hricik[7], Jocelyn Côté[8], Xiaosi Han[9], Wenping Zhou[10], Suparna Laha[11], Omar Abdel-Wahab[7], Ross L Levine[7], Glen Raffel[11], Yanyan Liu[10], Dongquan Chen[12], Haitao Li[4], Tim Townes[1], Hengbin Wang[1], Haiteng Deng[4], Y George Zheng[6], Christina Leslie[3], Minkui Luo[2], Xinyang Zhao[1*]

[1]Department of Biochemistry and Molecular Genetics, UAB Stem Cell Institute, The University of Alabama at Birmingham, Birmingham, United States; [2]Program of Molecular Pharmacology, Sloan Kettering Institute, Memorial Sloan Kettering Cancer Center, New York, United States; [3]Computational Biology Program, Sloan Kettering Institute, Memorial Sloan Kettering Cancer Center, New York, United States; [4]School of Life Sciences, Tsinghua University, Beijing, China; [5]Cell Signaling Technology, Inc., Danvers, United States; [6]Department of Pharmaceutical and Biomedical Sciences, The University of Georgia, Athens, United States; [7]Human Oncology and Pathogenesis Program, Sloan Kettering Institute, Memorial Sloan Kettering Cancer Center, New York, United States; [8]Department of Cellular and Molecular Medicine, University of Ottawa, Ottawa, Canada; [9]Department of Neurology, Comprehensive Cancer Center, The University of Alabama at Birmingham, Birmingham, United States; [10]Department of Internal Medicine, Zhengzhou - Henan Cancer Hospital, Zhengzhou, China; [11]Division of Hematology and Oncology, University of Massachusetts Medical School, Worcester, United States; [12]Division of Preventive Medicine, The University of Alabama at Birmingham, Birmingham, United States

*For correspondence: zhaox88@uab.edu

†These authors contributed equally to this work

Competing interests: The authors declare that no competing interests exist.

**Abstract** RBM15, an RNA binding protein, determines cell-fate specification of many tissues including blood. We demonstrate that RBM15 is methylated by protein arginine methyltransferase 1 (PRMT1) at residue R578, leading to its degradation via ubiquitylation by an E3 ligase (CNOT4). Overexpression of PRMT1 in acute megakaryocytic leukemia cell lines blocks megakaryocyte terminal differentiation by downregulation of RBM15 protein level. Restoring RBM15 protein level rescues megakaryocyte terminal differentiation blocked by PRMT1 overexpression. At the molecular level, RBM15 binds to pre-messenger RNA intronic regions of genes important for megakaryopoiesis such as GATA1, RUNX1, TAL1 and c-MPL. Furthermore, preferential binding of RBM15 to specific intronic regions recruits the splicing factor SF3B1 to the same sites for alternative splicing. Therefore, PRMT1 regulates alternative RNA splicing via reducing RBM15 protein concentration. Targeting PRMT1 may be a curative therapy to restore megakaryocyte differentiation for acute megakaryocytic leukemia.

**eLife digest** The many different cell types in an adult animal all develop from a single fertilized egg. The development of cells into more specialized cell types is called 'differentiation'. Proteins and other molecules from both inside and outside of the cells regulate the differentiation process.

RNA is a molecule that is similar to DNA, and performs several important roles inside cells. Perhaps most importantly, RNA molecules act as messengers and carry genetic instructions during gene expression. RBM15 is an RNA-binding protein that is found throughout nature, and is involved in a number of developmental processes. Previous research has linked the incorrect control of RBM15 with an increased risk of certain cancers, including megakaryocytic leukemia. However, it is not clear what role RNA-binding proteins such as RBM15 play during differentiation.

Now, Zhang, Tran, Su et al. have investigated the role of RBM15 during the development of large cells found in human bone marrow (called megakaryocytes). First, the experiments demonstrated that an enzyme called PRMT1 modifies RBM15. This enzyme adds a chemical mark called a methyl group at a specific site (an arginine amino acid) on the RNA-binding protein. Next, Zhang, Tran, Su et al. showed that the addition of this methyl group earmarks RBM15 for destruction. This means that an increase in PRMT1 levels reduces the amount of RBM15 in cells, while decreases in PRMT1 have the opposite effect.

Further experiments showed that RBM15 normally processes the RNA messengers that carry the genetic instructions needed for the differentiation of bone marrow cells. An excess of PRMT1 enzyme leads to a lack of this RNA-binding protein. This in turn interferes with the differentiation process, and can contribute to the development of cancers such as megakaryocytic leukemia. Future work will therefore explore whether targeting PRMT1 with drugs could represent an effective treatment for these kinds of cancers.

## Introduction

RNA binding proteins control post-transcriptional processing such as alternative RNA splicing, polyadenylation and protein translation, which is a prevalent part of gene regulation in normal cell differentiation and in cancer development (*Cabezas-Wallscheid et al., 2014*; *Chen et al., 2014*; *de Klerk and Hoen, 2015*; *Shapiro et al., 2011*). Arginine methylation of RNA binding proteins by protein arginine methyltransferases (PRMTs) regulates RNA splicing (*Bedford and Clarke, 2009*; *Bezzi et al., 2013*; *Cheng et al., 2007*; *Sinha et al., 2010*), subcellular localizations (*Matsumoto et al., 2012*; *Nichols et al., 2000*; *Tradewell et al., 2012*) as well as the binding affinity to RNA molecules (*Rho et al., 2007*). Nevertheless, the role of arginine methylation in regulating protein stability remains unknown. Here we demonstrate that an RNA binding protein, RBM15, is methylated by PRMT1, which triggers its ubiquitylation by an E3 ligase CNOT4.

RBM15 belongs to the split ends (spen) family of proteins. Spen proteins are evolutionarily conserved RNA binding proteins, which are involved in transcriptional regulation of Notch, Wnt and mitogen-activated protein kinase signal transduction pathways (*Chang et al., 2008*; *Chen and Rebay, 2000*; *Oswald et al., 2002*; *Rebay et al., 2000*; *Su et al., 2015*). Recently SPEN and RBM15 have been shown to be essential for Xist-mediated X chromosome inactivation (*Chu et al., 2015*; *McHugh et al., 2015*; *Minajigi et al., 2015*; *Moindrot et al., 2015*; *Monfort et al., 2015*). Genetic studies in *Drosophila* have shown that *spen* is required for cell-fate decision during development (*Kolodziej et al., 1995*). *FPA*, the *RBM15* homolog in *Arabidopsis,* controls flowering via regulating alternative polyadenylation of antisense RNAs at the *FLC* locus (*Hornyik et al., 2010*). RBM15 is essential for the development of multiple tissues in mouse knockout models, in particular, for the maintenance of the homeostasis of long-term hematopoietic stem cells and for megakaryocyte (MK) and B cell differentiation (*Niu et al., 2009*; *Raffel et al., 2009*; *Xiao et al., 2015*). Furthermore, RBM15 is involved in the chromosome translocation t(1;22), which produces the RBM15-MKL1 fusion protein associated with acute megakaryoblastic leukemia (AMKL) (*Ma et al., 2001*; *Mercher et al., 2001*).

Spen proteins consist of two domains: an RNA binding domain and a Spen Paralog and Ortholog C-terminal (SPOC) domain. Previously, spen proteins such as RBM15 and SHARP have been shown

to use the SPOC domains to recruit histone deacetylases for transcriptional regulation of Notch pathway and steroid receptor-dependent transcriptional regulation, and recruit mixed lineage leukemia (MLL) complexes to promoters for histone H3K4 methylation (*Ariyoshi and Schwabe, 2003*; *Lee and Skalnik, 2012*; *Ma et al., 2007*; *Oswald et al., 2002*; *Shi et al., 2001*; *Xiao et al., 2015*). Additionally, RBM15 is also involved in RNA export (*Uranishi et al., 2009*; *Zolotukhin et al., 2008*; *Zolotukhin et al., 2009*). RBM15 resides mainly within nuclear RNA splicing speckles by confocal microscopy (*Horiuchi et al., 2013*), suggesting that RBM15 is involved in RNA splicing. However, how spen proteins control cell differentiation is not described at molecular level.

In this report, we linked cellular differentiation to RBM15-regulated RNA metabolism using MK differentiation as a model. We demonstrated that RBM15 binds to specific introns of pre-messenger RNA (mRNA) of genes such as *RUNX1, GATA1* and *TPOR* (aka *c-MPL* or *MPL*), which play critical roles in MK differentiation, and to 3′UTRs of genes involved in RNA splicing and metabolic regulation. Reducing RBM15 protein concentration by PRMT1-mediated methylation favors the production of the alternatively spliced isoforms: RUNX1a, GATA1s and c-MPL-exon9-, a truncated c-MPL isoform. We also found that RBM15 promotes MK maturation in human primary cells. Therefore, PRMT1-RBM15 pathway fine-tunes cell differentiation via controlling the RBM15 protein concentration.

## Results

### RBM15 is methylated by PRMT1

In AMKL, MK differentiation is blocked. Gene expression data from AMKL patient samples (*Bourquin et al., 2006*) shows a higher PRMT1 expression level than other types of acute myeloid leukemia. Furthermore, high expression of PRMT1 is correlated with poor survival rate in acute myeloid leukemia (*Figure 1—figure supplement 1*). These clinical data strongly support that PRMT1 might be a key player in leukemogenesis. We applied bio-orthogonal profiling of protein methylation (BPPM) technology (*Figure 1—figure supplement 2*) (*Wang et al., 2011*) to identify proteins methylated by PRMT1 in MKs. We found that RBM15 is methylated at R578 by mass spectrometry analysis (*Figure 1—figure supplement 3A*). Alignment of the RBM15 sequences covering the methylation site shows that the methylation site is conserved across diverse species (*Figure 1A*) and downstream of the RBM15 RNA binding domains (*Figure 1—figure supplement 3B*).

We validated that two generic antibodies against mono-methyl arginine proteins and against di-methyl arginine proteins (Cell Signaling Inc.), can recognize the methylated peptides from RBM15 (NP_073605) R578 region, specifically in dot blots (*Figure 1—figure supplement 3C&D*). We observed that affinity purified Flag-tagged RBM15 protein is more efficiently methylated than the R578K mutant in western blots (WBs) in transfected 293T cells (*Figure 1B*, compare lanes 1 and 2) as well as in leukemia cells (*Figure 1—figure supplement 3E*). Given that the R578 is the last arginine in the RDRDRD repeat, we mutated all the three arginines, but still detected low methylation background, indicating other arginine residues not in the repeat are methylated (*Figure 1—figure supplement 3F*).

To determine whether PRMT1 is responsible, we overexpressed the two major isoforms of PRMT1 (V1 i.e.Q99873-3, V2 i.e. Q99783-1) in 293T cells together with wild type (WT) RBM15 (*Figure 1C*). Overexpressing either isoform enhanced the mono-methylation of RBM15 (*Figure 1C*, compare lanes 1, 2 and 3) and the V2 dimethylated RBM15 more effectively than the V1. Given that the V2 contains additional amino acids in the N terminal region relative to V1 (*Figure 1D*), it is striking that the PRMT1 V2 is primarily responsible for the dimethylation of RBM15 in vivo (*Figure 1C*).

In vitro methylation assays showed that the RBM15 peptide was dimethylated as efficiently as the histone H4 peptide without monomethylated peptides detected (*Figure 1—figure supplement 3G*). We then used purified Flag-RBM15 from 293T cells to conduct in vitro methylation assays. Only the WT RBM15 protein but not the R578K mutant was effectively dimethylated by PRMT1 in vitro (*Figure 1E*). These results together with in vivo methylation data collectively argue that mono-methylated RBM15 is a transient intermediate in vivo.

To confirm that PRMT1 is responsible for in vivo methylation, we made MEG-01 stable cell lines expressing doxycycline-inducible shRNA against PRMT1. We detected the methylated form of RBM15 with both mono-methyl and dimethyl antibodies. Reducing the expression of PRMT1

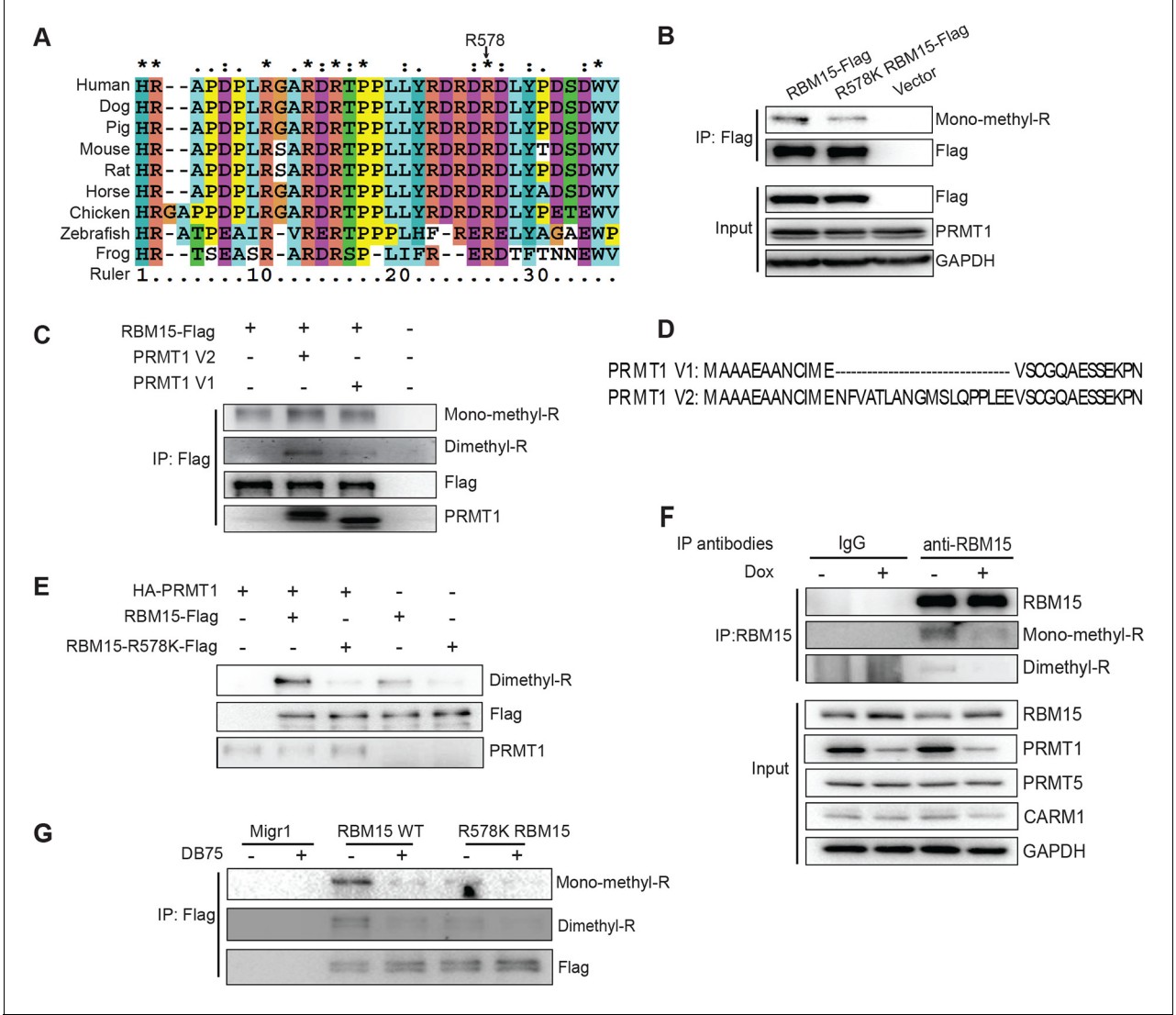

**Figure 1.** RBM15 is methylated by PRMT1 at R578 in mammalian cells. (**A**) Alignment of RBM15 sequences among different species shows R578 within a conserved protein region. (**B**) RBM15-Flag and its mutant (R578K), affinity purified with anti-Flag antibody from transfected 293T cells, were detected by WB with anti-monomethyl arginine antibody. (**C**) Flag-tagged RBM15 was affinity purified by Flag antibody for WB with two generic antibodies against mono-methyl arginine and dimethyl arginine. The 293T cells overexpressing wild type RBM15-Flag protein with PRMT1 V2 (lane 2) and V1 (lane 3) were treated with 20nM MG132 for 6 hr before harvesting. (**D**) The differences between N terminal sequences of PRMT1 V1 and V2 isoforms. (**E**) In vitro methylation assays. Affinity purified RBM15 protein was methylated by incubation with purified HA-PRMT1 and 0.15 mM of S-adenosyl-methionine at 30°C for 4 hours. The methylated RBM15 was detected by anti-dimethyl arginine antibody in WB. (**F**) RBM15 was immunoprecipitated with a mouse monoclonal anti-RBM15 antibody from whole cell extract prepared from a MEG-01 stable cell line expressing inducible short hairpin RNA against PRMT1. Normal IgG was used as immunoprecipitation controls. The immunoprecipitated RBM15 was detected by WB with antibodies against mono-methyl arginine (mono-R100) and asymmetrical dimethyl arginine (D4H5). As controls, we did WB with anti-PRMT5 and anti-PRMT4 antibodies. (**G**) Detection of methylated RBM15 in MEG-01-stable cell lines expressing Flag-RBM15 and R578K mutant proteins after a PRMT1 inhibitor (DB75) treatment for 24 hr. RBM15 is affinity purified by anti-Flag antibody and detected by WB with mono-methyl arginine antibody and dimethyl arginine antibody. PRMT, protein arginine methyltransferases; WB, western blot.

The following figure supplements are available for figure 1:

**Figure supplement 1.** PRMT1 is overexpressed in AMKL leukemia and associated with short survival rate in AML.

**Figure supplement 2.** RBM15 is discovered as a PRMT1 substrate via BPPM.

**Figure supplement 3.** Mapping the methylation site for RBM15.

*Figure 1 continued*

**Figure supplement 4.** RBM15 methylation status is further confirmed by a methyl-RBM15 antibody.

decreased the methylation level of the endogenous RBM15 (*Figure 1F*). When the leukemia cells were treated with DB75 (a PRMT1 inhibitor) (*Yan et al., 2014*), RBM15 methylation was reduced (*Figure 1G*). The methylation of RBM15 was further confirmed by an antibody raised against dimethylated RBM15 at R578 (*Figure 1—figure supplement 4*). Therefore, we concluded that PRMT1 enzymatic activity is responsible for the methylation of RBM15 at R578. RBM15 might be methylated by other PRMTs and at other sites since some methylation signals, especially the monomethylation signals, are present despite PRMT1 knock-down or inhibition and in the R578K mutant. Nevertheless, the PRMT1-dependent R578 methylation forms the majority of dimethylated species.

## RBM15 protein stability is determined by its methylation status

When RBM15 was co-expressed together with PRMT1, RBM15 protein level was reduced. Strikingly, the V2 isoform (detected by V2-specific antibody) caused a greater reduction of RBM15 protein level (*Figure 2A*), which coincided with the PRMT1 V2's capability for RBM15 dimethylation (*Figure 1C*). To further confirm that methylation triggers protein degradation, we used non-specific methyltransferase inhibitors (adenosine dialdehyde [adox] + methylthioadenosine [MTA]) (*Figure 2B*, left panel) as well as DB75 (*Figure 2B* right panel) to treat MEG-01 cells. Consistently, PRMT1 knockdown increased RBM15 protein level (*Figure 2C*, left panel) but not the RBM15 mRNA level (*Figure 2C*, right panel) in MEG-01 cells as well as in other leukemia cells (data not shown). Conversely, overexpression of PRMT1 V2 (*Figure 2D*, left panel) reduced RBM15 protein level. Given that the RBM15 mRNA levels in PRMT1 knockdown cells and in PRMT1 overexpressing cells were the same as in the control MEG-01 cells (*Figure 2D* middle and right panels), we concluded that PRMT1 controls RBM15 at protein level. Finally, we observed that the specific knockdown of PRMT1 V2 resulted in higher RBM15 protein level (*Figure 2E*). Taken together, we concluded that the PRMT1 V2 enzymatic activity plays a major role in regulating the stability of the RBM15 protein.

When we co-expressed RBM15 R578K together with PRMT1 V2 protein in 293T cells, we found that the mutant protein remained stable regardless of overexpression of PRMT1 V2 (*Figure 2F*, compare lanes 4 and 2). Thus, methylation of RBM15 at R578 is the direct cause for protein degradation. To further determine whether RBM15 degradation is affected by methylation on R578, we compared the half-life of the WT versus the mutant RBM15 proteins. The Flag-tagged RBM15 mutant has longer half-life compared with the WT RBM15 (*Figure 2G*). Tag does not affect protein stability as the tagged RBM15 and endogenous RBM15 have similar half-life.

## RBM15 degradation is an ubiquitylation-mediated process

We showed that the proteasome inhibitor MG132 stabilized the RBM15 protein, which indicates that the degradation is mediated by ubiquitylation (*Figure 3A*). To further confirm that the RBM15 protein is ubiquitylated, we used nickel beads to purify the poly-ubiquitylated form of RBM15 under denaturing conditions from 293T cells expressing the poly-histidine-tagged ubiquitin and Flag-tagged RBM15 (*Figure 3B*).

To investigate whether methylation of RBM15 by PRMT1 is required for RBM15 ubiquitylation, we mutated the methylation site to lysine. Compared with the WT RBM15 protein, significantly less R578K RBM15 protein was modified by ubiquitin (*Figure 3C* compare lanes 4 and 5,) when the respective plasmids were co-expressed in 293T cells. Thus, methylation on R578 for protein ubiquitylation is one major pathway for RBM15 degradation. To validate that endogenous RBM15 is also ubiquitylated in a methylation-dependent manner, we treated the MEG-01 cells with PRMT1 inhibitor DB75. After DB75 treatment, less ubiquitylated RBM15 proteins were observed (*Figure 3D*).

## CNOT4 is the E3 ligase responsible for RBM15 ubiquitylation

PRMT1 has been shown to physically interact with the CCR4-NOT complex via the CAF1 (aka CNOT7) subunit for RNA export (*Kerr et al., 2011*; *Robin-Lespinasse et al., 2007*). Furthermore, CNOT4, a subunit of the CCR4-NOT complex, has been shown to be an E3 ligase (*Albert et al.,*

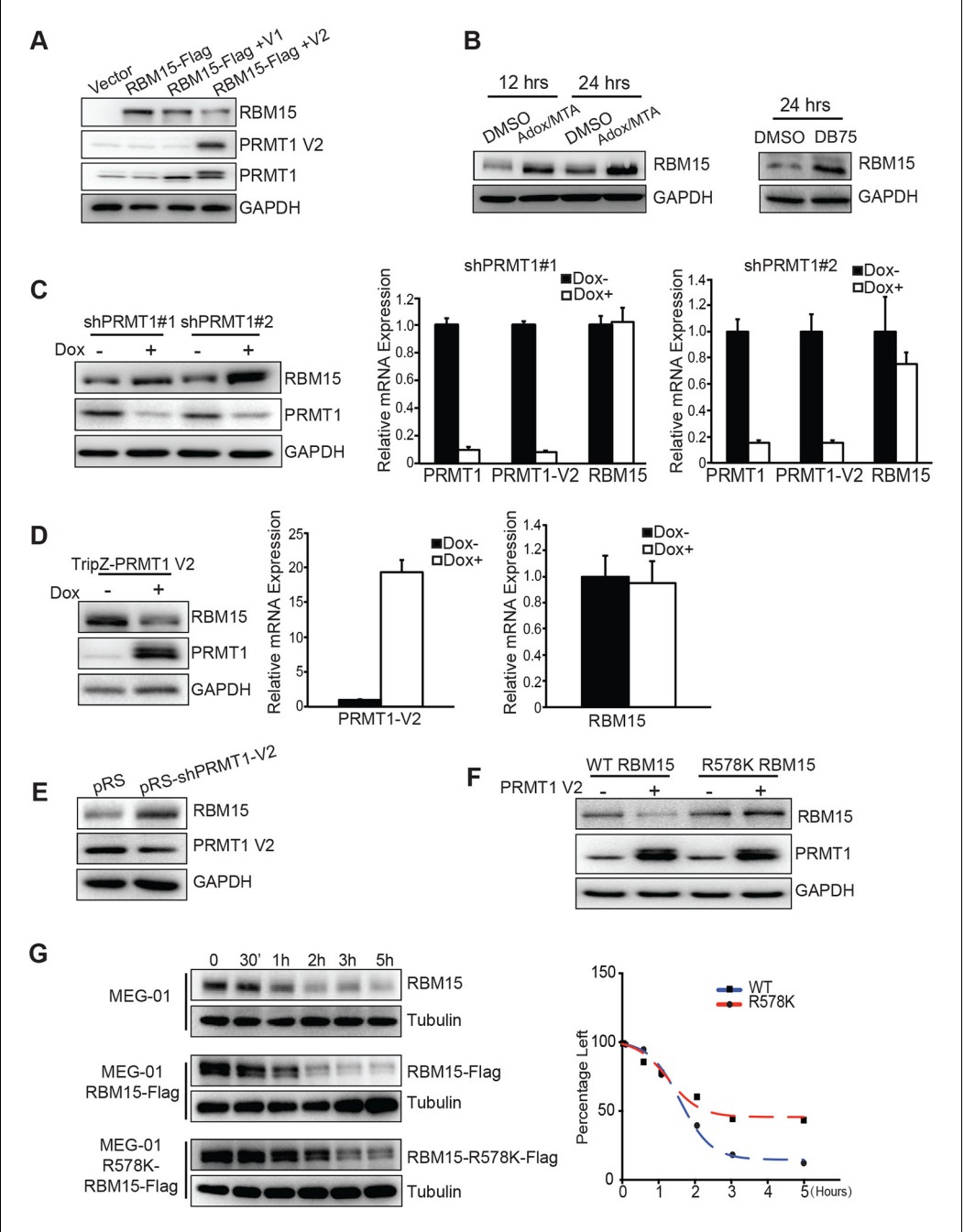

**Figure 2.** PRMT1 V2 isoform destabilizes RBM15 via methylation. (**A**) WB for RBM15 in 293T cells overexpressing PRMT1 V1 and V2. PRMT1 V2 was detected by anti-V2 specific antibody (PRMT1 V2). PRMT1 V1 and V2 were detected by an antibody for all isoforms (labeled as PRMT1). (**B**) The level of the RBM15 protein as detected by WB in MEG-01 cells treated with methyltransferase inhibitors (Adox and MTA mix) or DB75. (**C**) RBM15 protein level was measured by WB in MEG-01 cells with two doxycycline-inducible shRNA against PRMT1 (on the left). In the middle and right sides are real-time PCR results to show the mRNA levels of total amount of PRMT1, PRMT1 V2, and RBM15 in shPRMT1#1 stable MEG-01 cell line (middle) and in shPRMT1#2 stable MEG-01 cells (right). All data are presented as mean ± standard deviation from three independent experiments. (**D**) RBM15 protein level was measured by WB in MEG-01 cells induced by Dox to express PRMT1 V2 isoform. On the right are the real-time PCR charts for PRMT1 V2 and RBM15 mRNA levels. Data are presented as mean ± standard deviation from three independent experiments. (**E**) RBM15 protein level was accessed by WB in a MEG-01 stable cell line expressing shRNA against V2. The names of antibodies are listed on right. The pRS vector retrovirus infected MEG-01 cells were used as control. (**F**) WB with anti-Flag antibody to detect the protein levels of RBM15 wild type

*Figure 2 continued on next page*

*Figure 2 continued*

and R578K mutant proteins in 293T cells overexpressing PRMT1 V2 and RBM15 proteins. (G) The half-life of the RBM15 proteins in MEG-01 cells, and stable cell lines overexpressing Flag-tagged RBM15 and RBM15 R578K were assessed by WB. Cyclohemixide were added to stop protein synthesis 30 min before harvesting cells as the 0 time point. The half-life curves were plotted by GraphPad Prism 6. Adox, adenosine dialdehyde; DMSO, dimethyl sulfoxide; Dox, doxycycline; GAPDH, glyceraldehyde-3-phosphate dehydrogenase; mRNA, messenger RNA; MTA, methylthioadenosine; PCR; polymerase chain reaction; PRMTs, protein arginine methyltransferases; shRNA; short hairpin RNA; WB, western blot.

*2002*). To examine whether CNOT4 might be responsible for ubiquitylation of RBM15, we co-expressed CNOT4 with RBM15 in 293T cells. Strikingly, when RBM15 was co-expressed with PRMT1 and CNOT4, RBM15 was more severely ubiquitylated than RBM15 expressed with PRMT1 or CNOT4 alone (*Figure 3E*). We also detected the interaction between PRMT1 and RBM15 WT and mutant proteins in 293T cells (*Figure 3E* lanes 3,4 and 7,8). Moreover, only the WT and not the mutant RBM15 interacted with HA-tagged CNOT4 (*Figure 3E* lane 3, 4). In *Figure 3E* and *Figure 3C*, we showed that the RBM15 R578K, which cannot be methylated by PRMT1, had significantly less ubiquitylated RBM15 protein. This data strongly suggest that methylation is required for subsequent ubiquitylation by the CNOT4 complex.

To further confirm that CNOT4 is the enzyme responsible for endogenous RBM15 ubiquitylation, we showed that CNOT4 knockdown (WB in *Figure 3F* and real-time polychromase chain reaction [PCR] in *Figure 3—figure supplement 2A*) increased the amount of endogenous RBM15 protein in MEG-01 cells (*Figure 3F*). When MEG-01 cells were treated with DB75, CNOT4 knockdown did not further stabilize the RBM15 protein (*Figure 3F*). These results confirm that methylation is required for the subsequent degradation of the RBM15 protein. We used the clustered regularly interspaced short palindromic repeat (CRISPR) technology to knockout one allele of CNOT4 gene in 293T cells. We also found that removal of one allele of CNOT4 stabilized the endogenous RBM15 protein levels (*Figure 3—figure supplement 2B&C*).

We then purified the RBM15 and CNOT4 proteins (*Figure 3—figure supplement 1*) to do in vitro ubiquitylaton assay. The methylated RBM15 protein (Flag-tagged) by PRMT1 was more efficiently ubiquitylated by CNOT4 than the purified RBM15 R578 in vitro (*Figure 3G* compare lane 5 to lane 3). Taken together, we demonstrated that the RBM15 protein stability is negatively regulated by PRMT1-mediated methylation at R578, and CNOT4 is an E3 ligase responsible for degrading the methylated RBM15. To understand further whether the methylation site on RBM15 is sufficient to bind to CNOT4 complex, we synthesized two RBM15 peptides covering the arginine methylation sites to pull-down CNOT4 from cell extract. The di-methylated peptide bound specifically to the CNOT4 protein (*Figure 3H*), which is agreeable with the coimmunprecipitation data in *Figure 3E* showing that CNOT4 only interacted with WT RBM15 proteins. This result implies that either CNOT4 or subunits in the CNOT4 complex directly interacts with RBM15 via the methylation mark.

## PRMT1 V2 inhibits MK maturation via a methylation-ubiquitylation switch on RBM15

PRMT1 RNA levels were measured in isolated mouse long-term hematopoietic stem cells (LT-HSC), short-term hematopoietic stem cells (ST-HSC), progenitor cells and terminally differentiated lineages. PRMT1 expression level was the lowest in LT-HSC and higher especially in myeloid progenitor cells (*Figure 4—figure supplement 1A–D*). Megakaryocyte-–erythrocyte progenitor (MEP) cells expressed the highest level of PRMT1. We also checked the expression of PRMT1 and RBM15 in human cells with HemaExplorer (*Bagger et al., 2013*). The expression profiles of PRMT1 and RBM15 in different lineages are shared between humans and mice (*Figure 4—figure supplement 1E–G*). The mRNA level of RBM15 does not change as dramatically as that of PRMT1 among different lineages, which implies that post-transcriptional modification such as arginine methylation of RBM15 might contribute to the regulation of RBM15 protein levels.

To understand the biological functions of PRMT1 in MK differentiation, we used phorbol myristate acetate (PMA) to stimulate MEG-01 cells into mature MK cells (*Ogura et al., 1988*). Upon PMA stimulation, we observed upregulation of the RBM15 protein within 12 hr and simultaneous down-

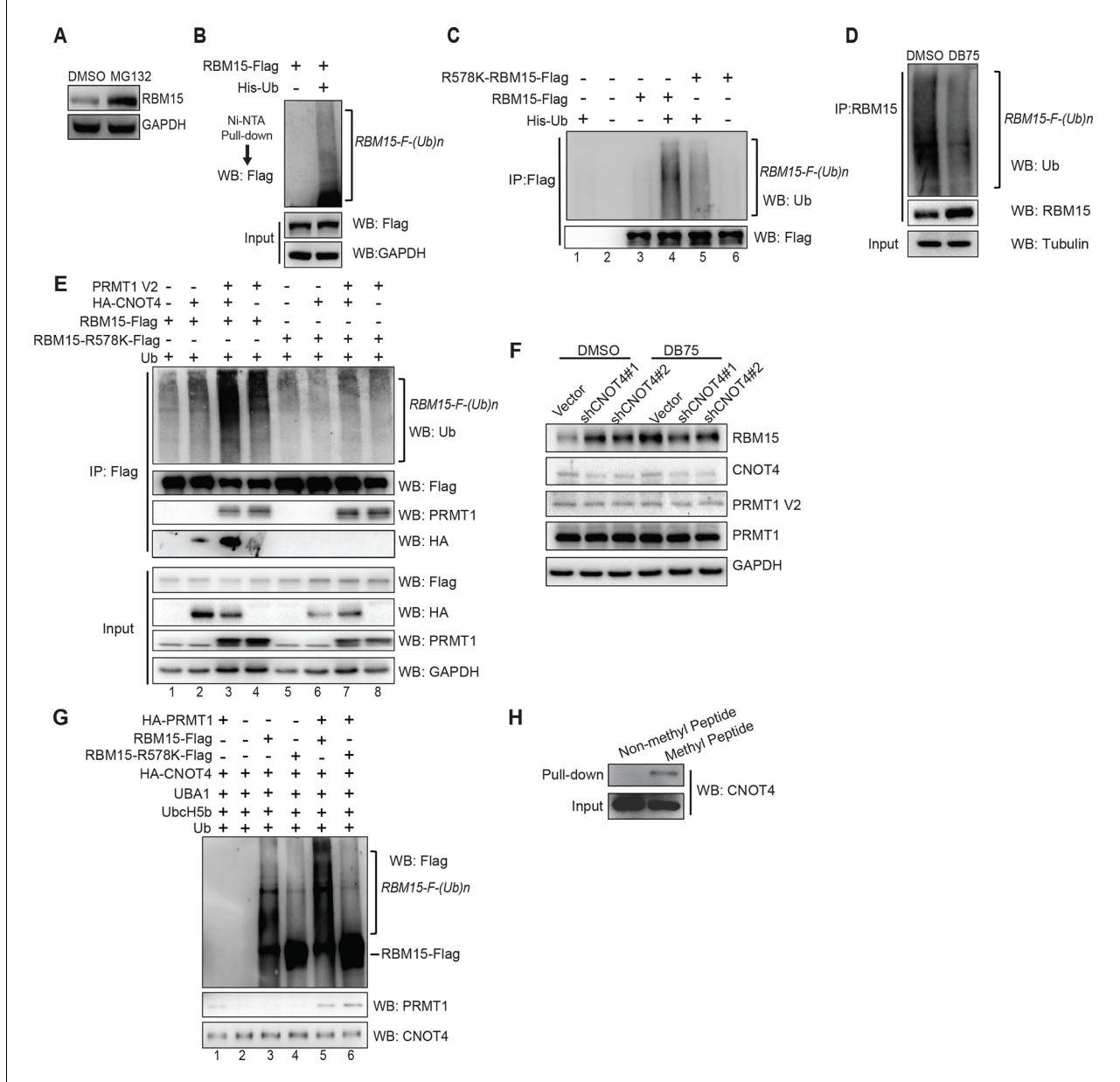

**Figure 3.** The RBM15 is ubiquitylated in a methylation-dependent manner. (**A**) WB for the RBM15 protein from MEG-01 cells treated with the proteasome inhibitor MG132. (**B**) The ubiquitylated RBM15-Flag was detected by anti-Flag antibody. The poly-ubiquitylated RBM15 was purified with nickel beads under denaturing conditions (6M of guanidine-HCl) from 293T cells expressing RBM15-Flag and poly-histidine-tagged ubiquitin. (**C**) The ubiquitylated RBM15 was measured by anti-ubiquitin antibody after affinity purified with Flag antibody from MG132 treated 293T cells expressing RBM15-Flag or R578K-Flag and ubiquitin. (**D**) IP-WB for poly-ubiquitylated RBM15 in DB75 treated MEG-01 cells. The endogenous RBM15 protein was immunoprecipitated by anti-RBM15 antibody and then blotted with anti-ubiquitin antibody and anti-RBM15 antibody. (**E**) The ubiquitylated RBM15 was detected by anti-Ub antibody after RBM15-Flag as well as its mutant was affinity purified from 293T cells transfected with combinations of plasmids shown above the gel. CNOT4 was detected via its HA tag. (**F**) WB to detect RBM15 protein levels in two MEG-01 cell lines expressing two different shCNOT4. PRMT1 inhibitor (DB75) was used to treat the cells expressing shCNOT4 RNAs. The efficiency of shCNOT4 knockdown was checked by real-time PCR. (**G**) In vitro ubiquitylation assays with CNOT4 and RBM15. Purified PRMT1 was added to methylate RBM15 in vitro in lanes 5 and 6 first before incubating with CNOT4 for in vitro ubiquitylation assays. All components were affinity purified from 293T cells. The ubiquitylated RBM15-Flag was detected by WB with anti-Flag antibody. (**H**) CNOT4 from MEG-01 whole cell extract was pulled down with methylated and nonmethylated peptides of RBM15. CNOT4 was detected by WB with anti-CNOT4 antibody. DMSO, dimethyl sulfoxide; GAPDH, glyceraldehyde-3-phosphate dehydrogenase; HA, hemagglutinin; IP, immunoprecipitation protocol; PCR, polymerase chain reaction; PRMTs, protein arginine methyltransferases; WB, western blot.

The following figure supplements are available for figure 3:

*Figure 3 continued on next page*

*Figure 3 continued*

**Figure supplement 1.** Purified proteins used in vitro methylation and ubiquitylation assays.

**Figure supplement 2.** The efficiency of CNOT4 knockdowns.

regulation of the PRMT1 V2 protein (*Figure 4A* left and middle panels). During the incubation period, neither the RBM15 mRNA level (*Figure 4A*, right panel) nor the total PRMT1 protein level (mainly V1 isoform) changed more than 1.5-folds, nevertheless we found that RBM15 protein level was significantly increased. The methyl-RBM15 level reduced coinciding with the decrease of the isoform V2 protein (detected by V2-specific antibody). Thus we conclude the isoform V2 –mediated methylation is one mechanism for upregulating RBM15 protein level, although we cannot rule out other possibilities such as enhanced protein translation. To assess RBM15 effects on MK differentiation, we ectopically expressed the WT and the R578K mutant proteins in MEG-01 cells by retroviruses. After stimulation with PMA for 7 days, more CD41$^+$ cells (*Figure 4B*) as well as higher levels of polyploidy (*Figure 4C*) were detected. In comparison to WT RBM15, the RBM15 R578K generated more CD41$^+$ cells, which may be explained by increased RBM15 R578K stability (*Figure 3*).

However, the cell line differentiation system can only mimic certain aspects of normal megakaryopoiesis. For example, we cannot detect CD42 in MEG-01 cells upon PMA stimulation. Therefore, we performed MK differentiation assays with adult human CD34$^+$ cells. Compared with the control, DB75-treated cells produced a higher percentage of CD61$^+$CD42$^+$ mature MK cells (*Figure 4D*). Conversely, ectopic expression of PRMT1 V2 in CD34$^+$ cells promoted the generation of single positive CD61$^+$ MK progenitor cells but blocked MK cell maturation (*Figure 4E*). Like PRMT1 overexpression, knockdown of RBM15 by shRNAs blocked MK maturation (*Figure 4F*, *Figure 4—figure supplement 2*), and overexpression of both RBM15 and its mutant promoted the maturation of CD34$^+$ cells (*Figure 4G*). Overexpression of RBM15 or RBM15 R578K rescued MK differentiation in CD34+ cells expressing PRMT1 V2 (*Figure 4H*). Taken together, PRMT1 controls megakaryopoiesis via controlling RBM15 protein levels.

## RBM15 binds to pre-mRNA of genes in MK differentiation

To understand how RBM15 protein level controls megakaryocytic differentiation, we affinity purified RBM15-associated proteins from 293T cells, which express endogenous RBM15 protein. Mass spectrometry analysis showed that RBM15 was associated with ASH2 and WTAP, two known RBM15-bound proteins (*Horiuchi et al., 2013*; *Lee and Skalnik, 2012*) as well as proteins involved in RNA splicing (*Figure 7—source data 1*), which is consistent with prior findings that RBM15 resides in nuclear splicing speckles (*Horiuchi et al., 2013*). Thus, we reasoned that RBM15 might regulate alternative splicing. We performed RNA immunoprecipitation assay (RIP) under stringent wash conditions with an RBM15 antibody recognizing the N-terminal region, followed by real-time polychromase chain reaction (RT-PCR) to detect genes known to be important for MK differentiation. We detected the bindings of RUNX1, GATA1 and c-MPL mRNAs with RBM15, while we did not detect that of control GAPDH mRNA (*Figure 5A*). Then we took an unbiased approach to identify global RBM15 target genes using RIP-seq. We have identified 1297 genes to which RBM15 binds directly (*Figure 5—source data 1*). Fifty-five percent of RBM15 binding sites are in intronic regions and 41% are in 3'UTR regions, with the remainder in 5'UTRs and CDS elements (*Figure 5B*). Gene ontology (GO) pathway analysis showed RBM15 regulates genes involved in differentiation and signal transduction via binding to intronic regions, whereas genes involved in RNA splicing and metabolic pathways were regulated via binding to 3'UTRs (*Figure 5C&D*). We found that overexpression of PRMT1 V1 and V2 isoforms as well as reducing the expression level of RBM15 enhanced mitochondria biogenesis in MEG-01 cells (*Figure 5—figure supplement 3*), which implies that PRMT1-RBM15 axis regulates metabolism. Among the RBM15 targeting genes with binding sites in introns are transcription factors known to be important for MK differentiation: *RUNX1, GATA1, and TAL1* (*Figure 5—figure supplement 1,2*). Although the transcription factor *LEF1* has not yet been linked to MK differentiation, LEF1 has been shown to interact with RUNX1 genetically and biochemically (*Daga et al., 1996*; *Mayall et al., 1997*; *McNerney et al., 2013*). RBM15 binding peaks on *c-MPL* pre-mRNA in the RIP-seq data (*Figure 5—figure supplement 2*).

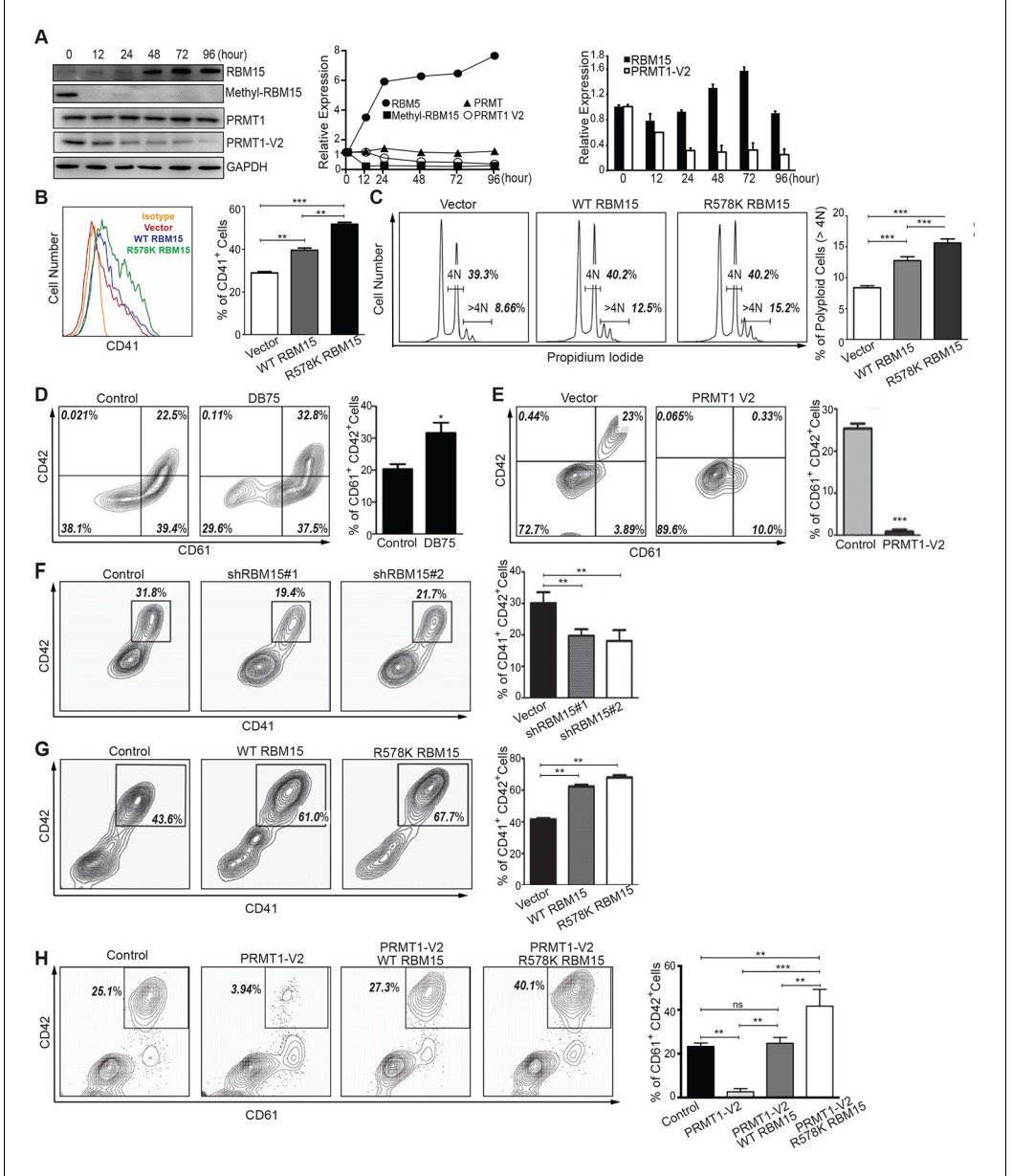

**Figure 4.** PRMT1 controls the protein level of RBM15 in MK maturation. (**A**) WB to measure the protein levels in MEG-01 cells stimulated to maturation by PMA. The left panel (WB results) showed the protein levels by antibodies against GAPDH, RBM15, methyl-RBM15, PRMT1 with a PRMT1 antibody against all isoforms and PRMT1 V2 with specific V2 antibody during the course of maturation. The middle panel shows the quantitation of the protein bands in the WBs on the left normalized to GAPDH. The right panel showed the decrease of PRMT1 V2 by real-time PCR during maturation with GAPDH mRNA as an internal control. Real-time PCR data were presented as means ± standard deviation from three independent experiments. (**B**) Histograms of CD41$^+$ cells on PMA-treated MEG-01 cells overexpressing RBM15 and RBM15R578K mutant proteins on day 3. The percentage of CD41$^+$ cells was calculated according to matched antibody isotype control. Three independent experiments were done with statistics shown on the left. P*** <0.001, P** <0.01. (**C**) FACS analysis of the polyploid status of PMA-treated cells overexpressing RBM15 and R578K mutant proteins by PI staining. Vector: lentivirus vector. P*** <0.001. (**D**) The matured MK cells were measured by CD61$^+$CD42$^+$. Human adult CD34$^+$ cells in pro-MK differentiation medium were treated with DB75 for 3 days. Three independent experiments were done with P*< 0.05. (**E**) Human adult CD34$^+$ cells were infected with lentivirus expressing PRMT1 V2 or lentivirus vector and grown in pro-MK differentiation medium for 5 days. Three independent experiments were done with P***<0.001. (**F**) Human adult CD34$^+$ cells were infected with two lentiviruses expressing shRNAs against RBM15 and grown in pro-MK differentiation medium. Three biological replicates were used for P value. P** <0.01. (**G**) Human adult CD34$^+$ cells were infected with lentiviruses expressing RBM15 or R578K proteins and grown in pro-MK differentiation medium for 5 days. Three independent experiments were done with P**<0.01. (**H**) Human adult CD34$^+$ cells were infected with lentiviruses expressing RBM15 or R578K together with a lentivirus expressing PRMT1 V2 and grown in pro-MK differentiation medium. Three biological replicates were used for P value. P**<0.01. GAPDH, glyceraldehyde-3-phosphate dehydrogenase; PI, propidium iodide; PMA, phorbol myristate acetate; PRMTs, protein arginine methyltransferases; WB, western blot; WT, wild type.

*Figure 4 continued on next page*

*Figure 4 continued*

The following figure supplements are available for figure 4:

**Figure supplement 1.** Relative expression levels of PRMT1 isoforms and RBM15 in different hematopoietic lineages derived from mouse and human.

**Figure supplement 2.** RBM15 protein levels in MEG-01 cell lines expressing two short hairpin RNA constructs against RBM15 by western blots.

To further investigate how RBM15 regulates alternative RNA splicing, we performed RNA-seq assays with RNA isolated from MEG-01 cells with or without RBM15 knockdown. The gene expression profile analysis showed that RBM15 knock-down alters metabolism and endoplasmic reticulum stress response pathways as well as proteins involved in chromatin assembly (*Figure 5—source data 2*). We then used two programs (*DEXSeq* and MISO) to detect the alterations of exon usage. *DEXSeq* detected 9704 differential exon usage events with P value lower than 0.05 and fold change higher than 1.2 fold. The MISO program detected 2027 differential exon usage events with Bayes factor higher than 2. RBM15 regulates differential exon usage in all eight categories in either directions (*Figure 5E,F*). Significant exon usage changes of 156 genes in the RIP-seq group were detected by both programs (*Figure 5—source data 3*) including transcription factors *RUNX1, GATA1, STAT5A, TAF9, TAL1, LEF1* and *ZNF160* as well as chromatin remodeling factors such as *macroH2A, DEK, BRD9, TLE3, NCOR1* and *HDAC4. RUNX1, GATA1, STAT5A* and *TAL1(SCL)* are well-studied transcription factors for their roles in MK differentiation (*Crispino, 2005*; *Olthof et al., 2008*; *Tijssen et al., 2011*). Signal transduction genes important for hematopoiesis such as *RPTOR* and *CDC42* (*Kaushansky and Kaushansky, 2014*) were found on the list as well. Therefore, RBM15 protein level may affect hematopoiesis via multiple pathways. At a molecular level, RBM15 not only affects alternative RNA splicing such as *RUNX1, GATA1, TAL1, STAT5A, LEF1* but also affects UTR utilization (e.g. *TAL1*), mixed exon usage (e.g. *CDC42*) and intron retention (e.g. *macroH2A*) (*Figure 5—figure supplement 4*). The MISO program was used to analyze how GATA1 is alternatively spliced. After knocking down RBM15, we found skipping of exon 2 on GATA1 (*Figure 5G*) and validated our results by resolving GATA1 isoforms in an agarose gel (*Figure 5H*).

## RBM15 regulates alternative splicing of genes important for MK differentiation

*GATA1* has two different isoforms: full-length GATA1 (GATA1fl), and short-form GATA1 (GATA1s), which is generated by skipping the exon 2 (*Rainis et al., 2003*). In Down syndrome leukemia, GATA1fl mRNA translates the GATA1s protein when mutations on GATA1fl mRNA create an alternative translation start site. Using isoform specific probes targeting to the exon junctions of GATA1s and GATA1fl to perform real-time PCR assays, we found that reducing RBM15 protein level favored the accumulation of GATA1s as the ratio of GATA1fl/GATA1s was reduced (*Figure 6A* top panel). Conversely, overexpression of RBM15 reversed the ratio of GATA1fl/GATA1s in favor of GATA1fl. The RBM15 R578K protein, which is more stable, altered the ratio more than the WT RBM15. Furthermore, overexpression of V2 in MEG-01 cells reduced the ratio of GATA1fl to GATA1s, similar to the effect of RBM15 knockdown. When we used inducible cell lines to knockdown PRMT1, we observed a higher ratio of GATA1fl/GATA1s-like in cells overexpressing RBM15 (*Figure 6A*). To further probe the importance of PRMT1 enzymatic activity in alternative splicing, we used PRMT1 inhibitor DB75 to treat MEG-01 cells, CMK cells and CMY cells, which all derived from AMKL leukemia patients. All three AMKL cell lines had higher GATA1fl/GATA1s ratio when treated with DB75 (*Figure 6—figure supplement 1*). Both isoforms of GATA1 can support MK differentiation, but only GATA1s supports unrestricted proliferation due to losing the interaction with E2F1 (*Li et al., 2000*). Therefore, fine-tuning the ratio between two different GATA1 isoforms regulates the balance between proliferation and differentiation of MK progenitors.

RUNX1 is required for MK differentiation as shown in a conditional RUNX1 knockout mouse model (*Ichikawa et al., 2004*). Individual RUNX1 isoforms regulate hematopoiesis differently (*Komeno et al., 2014*; *Tsuzuki and Seto, 2012*). RBM15 bound to RUNX1 pre-mRNA in intronic regions and in 3'UTR (*Figure 5—figure supplement 2*). The MISO program found that RUNX1 is alternatively spliced by skipping exon 6, which generates RUNX1-Exon 6⁻ protein. Overexpression of

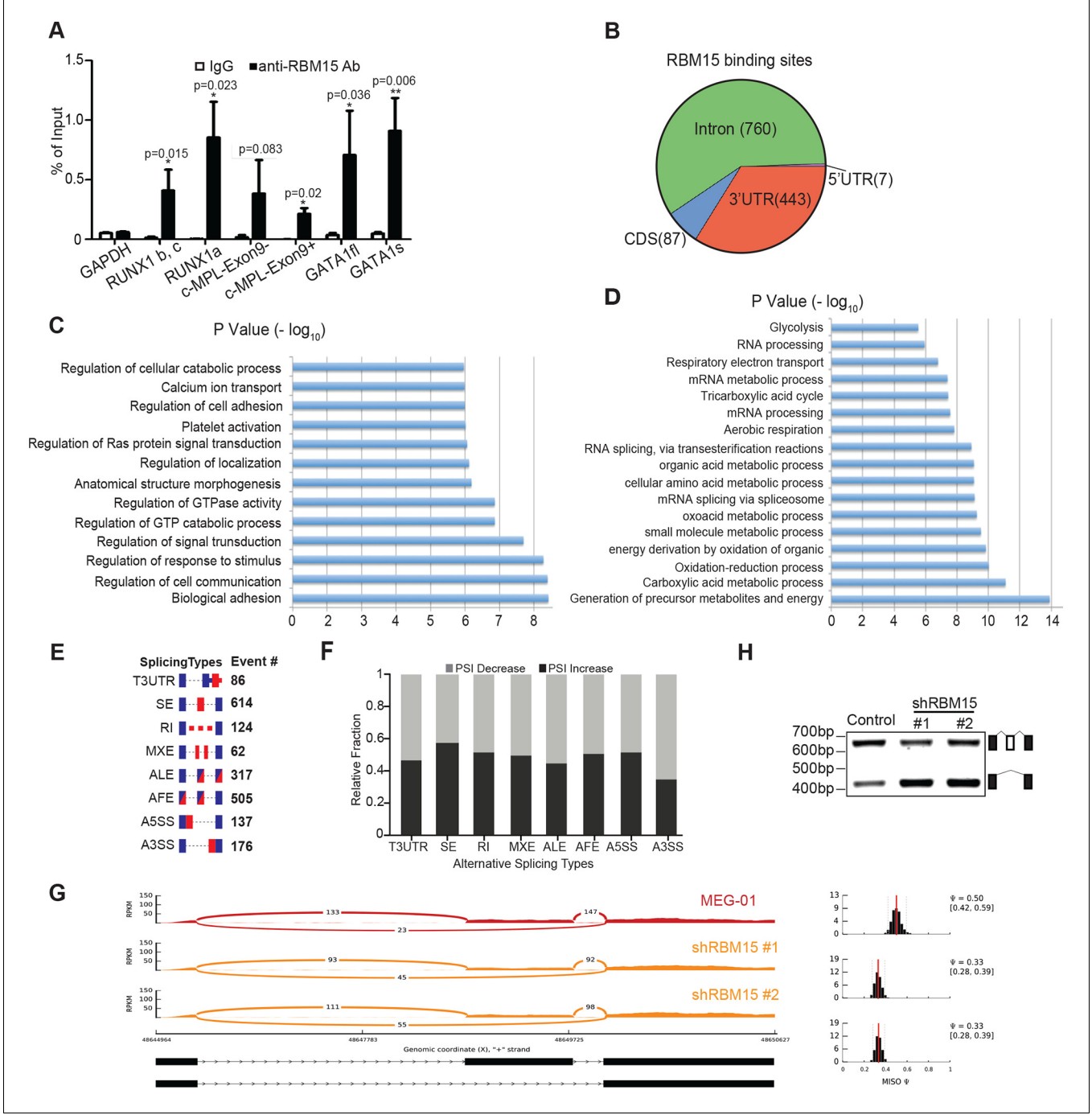

**Figure 5.** Analysis of RBM15 target genes. (**A**) Real-time PCR assays for detecting RNA associated with RBM15 in MEG-01 cells by RIP with the RBM15 antibody. The levels of RBM15-associated mRNAs were calculated as mean ± standard deviation from three independent experiments. (**B**) The distribution of RBM15 binding sites. All the RBM15 target genes were listed in *Figure 5—source data 2*. (**C**) GO pathway analysis (FDR<0.01) showed pathways associated with genes that have RBM15 binding sites in introns. (**D**) GO pathway analysis (FDR <0.01) revealed pathways associated with genes containing RBM15 binding sites in 3'UTR regions. (**E**) Differential exon usage events detected by the MISO program. (**F**) The changes of percentage splice-in events in different splicing categories when RBM15 is knocked down. (**G**) MISO plot for skipping of GATA1 exon 2 when RBM15 was knocked down. (**H**) Isoforms of GATA1fl and GATA1s were detected by PCR using RNA extracted from MEG-01 cells with or without RBM15 knockdown. ALE, alternative last exon; AFE, alternative first exon; A5SS, alternative 5' splicing sites; A3SS, alternative 3' splicing sites; GO, gene ontology; MXE, mutually exclusive exon usage; PCR, polymerase chain reaction; RI, retention intron; RIP, RNA immunoprecipitation assay; SE, skipped exon; T3UTR, tandem UTR.

*Figure 5 continued on next page*

*Figure 5 continued*

The following source data and figure supplements are available for figure 5:

**Source data 1.** Identification of RNAs associated with RBM15 by RNA immunoprecipitation assay with anti-RBM15 antibody.

**Source data 2.** Analysis of gene expression profile changes with RNA-seq data from RBM15 knockdown MEG-01 cells.

**Source data 3.** Analysis of differential exon usage regulated by RBM15 with RNA-seq data from RBM15 knockdown MEG-01 cells.

**Figure supplement 1.** RBM15 binding to pre-mRNA of genes known important for hematopoiesis.

**Figure supplement 2.** The RBM15 binding profiles on the c-MPL (**A**) and RUNX1 (**B**) pre-mRNAs.

**Figure supplement 3.** The mitochondria biogenesis is regulated by the PRMT1-RBM15 pathway.

**Figure supplement 4.** Representative genes detected by MISO and DEXSeq in the genes detected by RIP.

RUNX1-Exon 6⁻ in transgenic mice leads to reduced pool of hematopoietic stem cells (*Komeno et al., 2014*) (*Figure 5—figure supplement 4*). Given that RBM15 binds to the 3'UTR in exon 8 as well as to intronic regions flanking exon 7a, we investigated whether knockdown of RBM15 causes alternative 3'UTR usage to generate RUNX1a, which uses exon7a' 3'UTR. In RBM15 knock-down MEG-01 cells, the ratio between RUNX1a and RUNX1b,c switched in favor of RUNX1a. RUNX1a, which lacks the transactivation domain, acts as a dominant negative regulator to antagonize the function of RUNX1b,c (*Ran et al., 2013a*). Increasing the amount of RUNX1a may interfere with the differentiation into mature MKs. Consistent with RBM15 knockdown data, we showed that overexpression of RBM15 as well as knockdown of PRMT1 tilted the isoform balance toward RUNX1b,c. In agreement with the PRMT1 knockdown data, overexpression of PRMT1 V2 switched the ratio in favor of RUNX1a (*Figure 6A* middle panel).

The thrombopoietin receptor (TPOR, aka. c-MPL) is upregulated during MK differentiation. *c-MPL* has at least four described isoforms in humans (*Figure 6—figure supplement 2*). Using reported primers (*Li et al., 2000*) to amplify cDNA from MEG-01 cells, we found an additional band on the PAGE gel, which was identified as an undescribed isoform lacking exon 9 (c-MPL-exon9-–) through sequencing. The c-MPL-exon9– mRNA was predicted to yield a truncated protein (*Figure 6—figure supplement 2*). Thus we designed specific primer spanning the junction between exon 8 and 10 to detect c-MPL-exon9– mRNA. We also used a pair of primers, the forward primer spanning exon 8 and 9 and the reverse primer annealing to exon 9, to detect the full-length MPL (i.e. c-MPL-exon 9 +). The c-MPL-exon9-–/c-MPL-exon9+ ratio was higher in favor of c-MPL-exon9- (*Figure 6A*, bottom panel) when RBM15 protein level is reduced. Furthermore, when RBM15 or RBM15 R578K was overexpressed, the ratio was down in favor of c-MPL-exon9+. Given that c-MPL exon9+ is required for MK differentiation, the data further support for the positive role of RBM15 at MK maturation. All other isoforms with part of exon 9 or with both exon 9 and 10 missing were spliced in the same way by RBM15-PRMT1 pathway (*Figure 6—figure supplement 3*). Thus, RBM15 enhances the inclusion of exons to produce full-length functional c-MPL like in RUNX1's case. Consistent with our data, mouse *Rbm15* regulates c-Mpl alternative splicing in the same fashion (*Xiao et al., 2014*).

We monitored the isoform ratios using human adult CD34⁺ cells in pro-MK differentiation medium over a time course. We found that the GATA1fl/GATA1s ratio was increased, and the RUNX1a/RUNX1b,c ratio and c-MPL-exon9–/c-MPL-exon9+ ratio were decreased during the course of differentiation (*Figure 6B*). We observed that the isoform ratio changes in the DB75-treated CD34⁺ cells (*Figure 6C*) exactly like in the DB75-treated MEG-01 cells (*Figure 6—figure supplement 1*). Based on these data, we concluded that RBM15 regulates alternative RNA splicing of genes important for MK differentiation. By controlling the RBM15 protein dosage, PRMT1 thereby regulates MK differentiation.

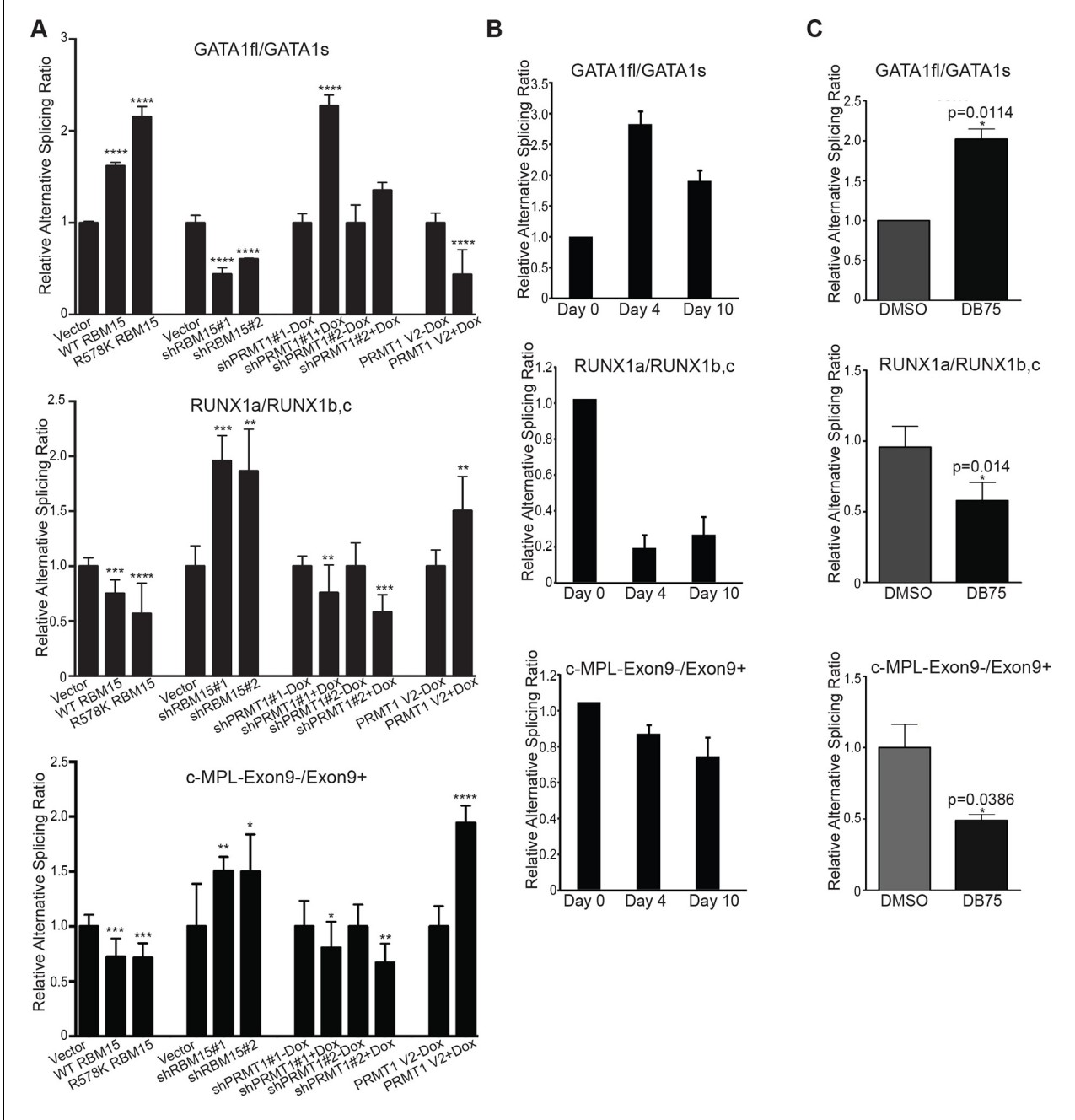

**Figure 6.** Methylation of RBM15 controls alternative splicing of genes (*RUNX1, GATA1* and *c-MPL*) important for MK differentiation. (**A**) Alternative splicing of RUNX1, GATA1, c-MPL in MEG-01 cells and MEG-01-derived stable cell lines overexpressing RBM15 and PRMT1 V2 or knocking down RBM15 and PRMT1 V2. The ratios of different isoforms were calculated from real-time PCR assays with isoform specific primers. At least three independent experiments were performed. P****: <0.01; P***: <0.05; P**: <0.2 and P*: <0.3 compared to their respective vector control groups. (**B**) Time course for alternative splicing of RUNX1, GATA1 and c-MPL in human adult CD34+ cells grown in pro-MK differentiation medium. Three independent experiments were used to calculate the standard deviation. (**C**) The alternative splicing of GATA1, RUNX1 and c-MPL was measured as ratio change in human adult CD34+ cells treated with DB75 overnight in basic cytokine mix. Three independent experiments were used to calculate the P values. PCR, polymerase chain reaction; PRMT, protein arginine methyltransferase

The following figure supplements are available for figure 6:

**Figure supplement 1.** Alternative splicing of GATA1 is regulated by PRMT1 in acute megakaryoblastic leukemia cell lines.

**Figure supplement 2.** Schematic diagram of MPL isoforms.

*Figure 6 continued on next page*

Figure 6 continued

**Figure supplement 3.** Alternative splicing of c-MPL is measured as the ratios of different isoforms by real-time PCR assays.

## RBM15 controls alternative splicing via its interaction with SF3B1

To understand the detailed mechanism of how RBM15 regulates alternative splicing, we identified RBM15-associated splicing factors such as SF3B1 and U2AF, known proteins in committed RNA splicing A complex that bind to branch point region (*Figure 7—source data 1*). Furthermore, we confirmed the interaction between SF3B1 and RBM15 by co-immunoprecipitation with anti-Flag antibody for Flag-tagged RBM15 proteins (*Figure 7A*) and with anti-RBM15 antibody (*Figure 7B*). To further confirm that PRMT1 enzymatic activity is important for alternative RNA splicing, we inhibited PRMT1 activity with DB75 or shPRMT1. The interaction between SF3B1 and RBM15 was subject to methylation regulation. One simple explanation is that RBM15 might recruit more SF3B1 because of more RBM15 and SF3B1 proteins available. It is also possible that methylation might be directly involved in RBM15 and SF3B1 interaction. Given that co-immunoprecipitation assays cannot distinguish the direct and indirect protein-–protein interaction in a protein complex, interaction of RBM15 with SF3B1 only implies that RBM15 recruits the 3'RNA splicing complex. Since SF3B1 is often mutated in leukemia cells, we sequenced the *SF3B1* gene in MEG-01 cells and confirmed that *SF3B1* was not mutated in this cell line. Thus the interaction is between two WT proteins.

From the RIP-seq data, we found that RBM15 was specifically associated with intronic region close to 5'end of intron 1 of GATA1 (*Figure 7C*). We validated the RIP-seq data by real-time PCR with primer sets covering the GATA1 intron 1. Analysis of immunoprecipitated RNA associated with SF3B1 indicated that SF3B1 also preferentially bound to the same region that RBM15 bound to. Furthermore, with RBM15 knockdown, less SF3B1 was associated with GATA1 pre-mRNA (*Figure 7D*). On the other hand, GAPDH has similar level of SF3B1 binding to introns regardless of RBM15 expression level. Thus, RBM15 is responsible for recruiting SF3B1 to pre-mRNA molecules. In this case, RBM15 facilitates the inclusion of GATA1 exon 2. For c-MPL pre-mRNA, RBM15 preferentially bound to the intron 8 and 10 of the pre-mRNA molecules with two distinct peaks (*Figure 7E* left), while SF3B1 (*Figure 7E* right) did bind to regions between the two RBM15 bound peaks at higher level than other regions. When we used RBM15 knockdown MEG-01 cells to do SF3B1 RIP on c-MPL pre-mRNA, we found that SF3B1 binding was significantly reduced in RBM15 bound regions (*Figure 7E* right). In this case, RBM15 functions as a splicing enhancer for inclusion of exons 9 and 10. In aggregate, these data show that RBM15 is responsible for recruiting SF3B1 containing branch point recognition complex to introns, and methylation controls the RBM15 dosage thereby controlling alternative RNA splicing (*Figure 7F*).

## Discussion

We report here that PRMT1 V2 dimethylates RBM15 within the sequence (LYRDRDR(me2)DLY). Although the most common sequence for arginine methylation is arginine flanked by glycines, PRMT1 has been previously reported to methylate arginines in PGC1-α and in RUNX1 flanked by other amino acids (*Teyssier et al., 2005*; *Zhao et al., 2008*). Recent proteomics work further proves that the methylation motif can be very diverse (*Guo et al., 2014*). Since we can still detect a low level of mono-methylation signal after R578K mutation, we believe that the mono-methylation signals come from additional arginines methylated by PRMT1 or other arginine methyltransferases. Protein mono-methylation sites, in general, are redundant and prevalent and are catalyzed by multiple protein arginine methyltransferases (*Dhar et al., 2013*). Nevertheless, these potential sites are not involved in methylation-mediated ubiquitylation. Another surprising result is that the V2 isoform, which contains additional amino acids on the N-terminus, primarily dimethylates RBM15 in vivo (*Figure 1C*). Data from PMA-treated MEG-01 cells argue that the V2 isoform is more relevant to MK maturation (*Figure 4A*). Although it has been demonstrated that V1 and V2 have different substrate specificities in vivo (*Goulet et al., 2007*), RBM15 is the first V2-specific substrate identified. Of note, the V2 isoform only comprises less than 5% of total PRMT1 level, but V2 is mainly responsible for oncogenic activities in colon and breast cancers (*Baldwin et al., 2012*; *Papadokostopoulou et al.,*

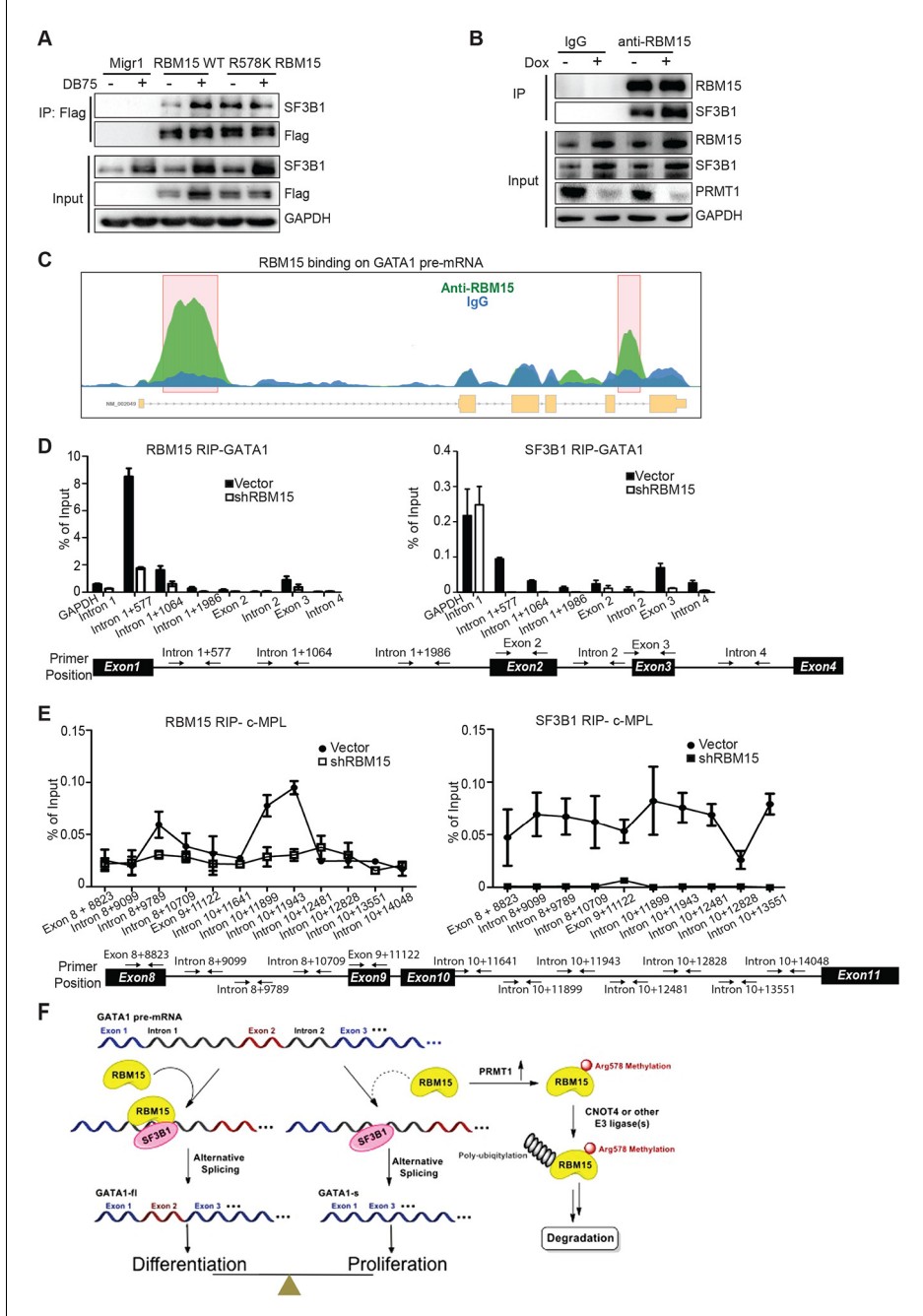

**Figure 7.** RBM15 directly recruits the intron-binding splicing factor, SF3B1, for alternative RNA splicing. (**A**) The interaction between SF3B1 and RBM15 in the context of PRMT1-mediated methylation. RBM15-Flag and RBM15 R578K-Flag expressed from two MEG-01 cell lines with or without DB75 treatment were immunoprecipitated with anti-Flag antibody for detecting interaction with SF3B1 by WB. (**B**) The endogenous SF3B1 was co-immunoprecipitiated with anti-RBM15 antibody in MEG-01 cells expressing inducible shPRMT1. Normal mouse serum was used as a negative control. (**C**) RBM15 binding profile on GATA1 pre-mRNA based on RIP-seq data. The green peaks are the binding sites for RBM15 and the blue profile is the binding profile for normal IgG. Two biological replicates were used for bioinformatic analysis. The significant peaks were shaded with pink squares. (**D**) The regions where RBM15 (RIP with RBM15 antibody, left panel) and SF3B1 (RIP with SF3B1 antibody right panel) bound on GATA1 pre-mRNA in MEG-01 cells (solid bar) and RBM15 knockdown MEG-01 cells (open bar) were mapped by real-time PCR assays. The locations of primers on the pre-mRNA of GATA1 were shown on the bottom. Three biological replicates were used to calculate the standard deviations. GAPDH intron 1 was used as negative controls for both antibodies. (**E**) The regions on c-MPL pre-mRNA, where RBM15 and SF3B1 bound in

*Figure 7 continued on next page*

*Figure 7 continued*

MEG-01 cell lines expressing shRBM15 (square line) or expressing pLKO vector (solid dot line), were assessed by RIP with RBM15 (left panel) and SF3B1 antibodies (right panel). The locations of primers on the pre-mRNA of c-MPL are shown on the bottom. Three biological replicates were used for standard deviations. (F) A model for RBM15-mediated regulation of alternative RNA splicing. RBM15 and SF3B1 cooperate to produce GATA1fl and low level of RBM15 leads to lower SF3B1 binding and skipping of the exon 2. PRMT1-mediated methylation of RBM15 controls the ubiquitylation of RBM15 by CNOT4, thus controlling the balance between proliferation and differentiation in megakaryopoiesis. GAPDH, glyceraldehyde-3-phosphate dehydrogenase; mRNA, messenger RNA; RIP, RNA immunoprecipitation assay; PCR, polymerase chain reaction; PRMT, protein arginine methyltransferase

The following source data is available for figure 7:

**Source data 1.** Mass spectrometry analysis of RBM15-associated proteins.

*2009*). We cannot completely exclude a role for V1 in RBM15 methylation due to the inability to knockdown V1 specifically. Because developing inhibitors specific for PRMT1 V2 isoform will potentially alleviate off-target effects, which remain a major impediment for developing anti-PRMT1 inhibitors, our data offers new clues on how to achieve V2-specific inhibition. RBM15 is involved in chromosome translocation t(1;22), which produces the fusion product RBM15-MKL1 (*Ma et al., 2001*; *Mercher et al., 2001*). Whether the fusion protein, which retains R578, is methylated by PRMT1 is still an unanswered question.

It was previously reported that methylation of a transcriptional coactivator p/CIP by PRMT4 or CARM1 leads to its degradation, however the mechanism was incomplete (*Naeem et al., 2007*). Here we report that PRMT1 methylation triggers protein degradation via ubiquitylation by the E3 ligase CNOT4. CNOT4 is loosely associated with the other subunits of CCR4-NOT complex (*Lau et al., 2009*). Apart from deadenylation activity of other CCR4-NOT complex, CNOT4 has been shown to ubiquitylate a histone demethylase Jhd2, which is responsible for erasing the histone H3K4 trimethylation marks in yeast (*Mersman et al., 2009*). Given that RBM15 interacts with histone methyltransferases complexes to maintain histone H3K4 methylation level (*Lee and Skalnik, 2012*; *Xiao et al., 2015*), our data provide new insights into a potential crosstalk between methyltransferases and demethylases.

Recently, the E3 ligase WWP2 has been shown to recognize the methylated lysine in SOX2 through its HECT domain (*Fang et al., 2014*). Given that CNOT4 is within a large CCR4-NOT complex with nine subunits, we have not determined which domain in CNOT4 or other subunits of CCR4-NOT complex recognizes the methylated RBM15. The peptide pull-down experiments strongly support that methylated region on RBM15 is sufficient for binding to CNOT4. Since we found that the RING domain on CNOT4 alone was not sufficient for binding to methylated RBM15 (data not shown), it is possible that more than one region on CNOT4 or other subunits in CCR4-NOT complex might interact with RBM15.

In order to investigate the biological significance of PRMT1-mediated methylation of RBM15, we analyzed the role of PRMT1-RBM15 axis in hematopoiesis. Normal MK differentiation can be divided into two phases: from hematopoietic stem cells to MK progenitor cells and from MK progenitor cells into mature polyploidy MK cells. Genes such as ARHGEF1 (ARHGEF1,2 are in *Figure 5—source data 1*) play opposite roles at different phases of MK differentiation (*Gao et al., 2012*; *Smith et al., 2012*). The number of MEP progenitor cells increases, while the polypoidy of mature MKs decreases in *Rbm15* knockout mice (*Niu et al., 2009*). Consistent with this, RBM15 knockdown in human primary cells like Rbm15 knockdown in mice produces low percentage of mature MKs (*Figure 4F*). PRMT1 promotes the production of MEP cells, but PRMT1 has to be turned off to generate mature, polyploid CD41$^+$/CD61$^+$CD42$^+$ MK cells. Because both PRMT1 and RBM15 are ubiquitously expressed in all tissues, PRMT1-–RBM15 might regulate differentiation in the same fashion in other tissues such as cardiac tissue, spleen and placenta where RBM15 has been shown to be essential for development (*Raffel et al., 2009*).

According to the Oncomine database (www.oncomine.com), PRMT1 is highly expressed in acute myeloid and lymphoid leukemia (AML and ALL) as well as in solid tumors (*Rhodes et al., 2007*).

PRMT1 accounts for 4% gene mutations, amplifications and deletions in solid tumors (see www.cbio-portal.org) (*Gao et al., 2013*). The relevance of PRMT1 inhibition in cancer therapies is summarized in reviews (*Greenblatt and Nimer, 2014*; *Yang and Bedford, 2013*). The roles of PRMT1 in supporting cancer cell proliferation are shown in AML1-ETO and MLL-EEN-initiated leukemia (*Cheung et al., 2007*; *Shia et al., 2012*). A PRMT1-mediated MK block may therefore be contributory to AMKL leukemogenesis. Apart from their functions in MK differentiation, RUNX1, GATA1 and c-MPL as well as other epigenetic factors such as BRD4 found in this study are dysregulated in a broad spectrum of hematological malignancies as well as solid tumors (*Baratta et al., 2015*; *Crawford et al., 2008*; *Crispino, 2005*; *Harada et al., 2004*; *Pikman et al., 2006*). Thus, dysregulation of PRMT1-–RBM15 pathway might be a common mechanism in leukemia and solid tumors.

The role of RNA splicing is increasingly appreciated in hematological malignancies (*Cazzola et al., 2013*). The RNA splicing patterns vary widely from hematopoietic stem cells to progenitor cells in humans (*Chen et al., 2014*). Our studies offer mechanisms on how RNA splicing is regulated during hematopoiesis. RBM15 directly recruits the SF3B1 branch point recognition complex to introns. *SF3B1* has been found to have mutations in myelodysplastic syndrome and in leukemia (*Yoshida et al., 2011*). Given that SF3B1 knockout mouse cannot mimic defects seen in myelodysplastic syndromes (*Wang et al., 2014*), mutations in *SF3B1* are likely gain-of-function mutations. Whether altered interactions between RBM15 and SF3B1 mutants contribute to hematological malignancies warrants future study.

One direct consequence of RNA splicing is the altered concentration ratios between functional proteins and their alternatively spliced dominant negative counterparts. The physiological significance for multiple isoforms of *c-MPL* is to attenuate signaling from full-length c-MPL during MK maturation. In transgenic mouse exclusively expressing full-length c-MPL, excessive production of platelets is observed (*Tiedt et al., 2009*). Abnormal expression of GATA1s and RUNX1a are associated with leukemia (*Crispino, 2005*; *Liu et al., 2009*). By controlling isoform ratios, HSCs fine-tune the lineage preference and the magnitude of lineage production during differentiation. This type of regulation is very important for HSCs to respond quickly to ever-changing environmental clues.

RBM15 by binding to introns flanking the alternatively spliced exons enhances the inclusion of the exons to produce full length c-MPL mRNA (*Figure 7E*). In the case of GATA1, RBM15 binds to intron 1 to recruit SF3B1 to the same location (*Figure 7D*). As a result, RBM15 enhances the inclusion of exon 2 to generate the GATA1fl mRNA (*Figure 6A*). RBM15 facilitates the production of more full-length RUNX1b,c by skipping the exon that encodes RUNX1a. Knockdown RBM15 also facilitate the generation of RUNX1-exon6-⁻. Based on our proteomic data, RBM15 binds to RNA binding proteins that are involved in RNA surveillance, transport and degradation (*Figure 7—source data 1*). Consistently, we also detected association of RBM15 with matured mRNAs (*Figure 5A*). In addition, RBM15 associates with transcriptional elongation factors such as Tho2 and CTR9 besides its role in transcription initiation by positively regulating histone H3K4 methylation (*Xiao et al., 2015*). We speculate that RBM15 participates in regulating the whole lifecycle of a messenger RNA. This is reminiscent of the functions of CCR4-NOT complex (*Miller and Reese, 2012*). RBM15 has been shown to be a part of RBM20-mediated RNA splicing network in cardiomyocytes (*Maatz et al., 2014*; *Raffel et al., 2009*). Thus, RBM15 is involved in the development of other tissues in addition to hematopoiesis. Further experiments to define the specificity and affinity of RBM15 binding sequences will help us to understand the mechanisms of alternative splicing mediated by RBM15.

Apart from binding to introns, RBM15 binds to 3′UTR of metabolic enzymes, splicing factors as well as a few long noncoding RNA molecules and microRNAs (*Figure 5—source data 2*). Given that splicing factors can be involved in alternative RNA polyadenylation and RNA export, it is not surprising that RBM15 binds to 3′UTRs for alternative RNA polyadenylation, export as well as for RNA splicing (*Castelo-Branco et al., 2004*; *Kyburz et al., 2006*). Analysis by the MISO program indicates that RBM15 regulates alternative RNA splicing of genes such as HK1 by binding to 3′UTR. In the future, deeper sequencing of RNA samples and performing high-resolution PAR-CLIP assays (*Hafner et al., 2010*) with RBM15 antibodies will further define the RBM15-mediated RNA metabolism.

In summary, our findings demonstrated a new role for PRMT1 in regulating RNA metabolism. Given that both RBM15 and PRMT1 are evolutionarily conserved proteins from plants to mammals, our studies on the PRMT1-–RBM15 pathway offer mechanistic insights into how *RBM15* may regulate cell fate decision in multi-cellular organisms at post-transcriptional levels.

## Materials and methods

### Using BPPM technology to profile PRMT1 methylated proteins

To reveal proteome-wide substrates of protein methyltransferases such as PRMT1 in the context of complex cellular components, we took advantage of the emerging BPPM technology (*Luo, 2012*; *Wang and Luo, 2013*). In BPPM, designated methyltransferases are engineered to process sulfonium-alkyl SAM analogues as alternative cofactors and thus transfer the distinct sulfonium alkyl handles to substrates (*Figure 1—figure supplement 2A*). Because the alkyl handles contain a terminal-alkyne for the azide-alkyne Huisgen cycloaddition (the click reaction), we can then couple the chemical moiety with a biotin-containing azide probe for amenable target enrichment and characterization. With the previously-identified Y39F-M48G PRMT1 mutant and the matched 4-propargyloxy-but-2-enyl SAM (Pob-SAM) as BPPM reagents (*Wang et al., 2011*), RBM15 in the context of Meg-01 cells was readily labeled by the enzyme-–cofactor pair, modified by a biotin-containing azide probe, and enriched by streptavidin beads (*Figure 1—figure supplement 2B*). As the negative control, the label efficiency dropped significantly in the absence of the Y39F-M48G PRMT1 variant. The tagged proteins were subsequently purified with streptavidin beads for mass spectrometry analysis. RBM15 was identified among the methylated proteins. We verified the mass spectrometry data with direct WB with RBM15 antibody against biotinylated proteins (*Figure 1—figure supplement 2C*). To further map the RBM15 methylation sites, Flag-tagged RBM15 protein was purified from transient transfected 293T cells for tandem mass spectrometry analysis. A mono-methylation site was found at R578, which lies between the RNA-binding domain and the SPOC domain (*Figure 1—figure supplement 3A*).

### Plasmids

The human RBM15 cDNA in pCDNA3 vector was kindly provided by Dr. Barbara Felber (NIH). The nucleotide sequences of all constructs were verified by sequencing. The retrovirus expression vector Migr1 was used to overexpress RBM15 and its mutant. pTripZ (Openbiosystems) was used to overexpress the PRMT1-V1 and PRMT1-V2 proteins by replacing RFP with PRMT1-V1 or PRMT1-V2 cDNA. The lentivirus expressing shRNA against PRMT1, RBM15, and CNOT4 were purchased from Openbiosystems (ThermoScientific Inc.). The sequences are available in *Supplementary file 1*. pRS and pRS-shPRMT1-V2 plasmids were reported previously (*Baldwin et al., 2012*).

### Viral production and transduction

For virus production, 293T cells were cotransfected with retroviral or lentiviral plasmids with helper plasmids as described (*Vu et al., 2013*). Cells stably expressing PRMT1-V1 and PRMT1-V2 or knockdown RBM15 or PRMT1 were selected with 5 µg/ml puromycin, and GFP-positive cells stably expressing Flag-RBM15 and Flag-RBM15-R were sorted on a FACSAria cell sorter. BD FACS LSRFortessa was used to do FACS analysis.

### New isoform of c-MPL

The gene bank number for this new c-MPL isoform is KF964490.

### Cell culture

HEK293T cells were cultured in DMEM supplemented with 10% FBS and transient transfected by lipofectamine 2000 (Invitrogen, Grand Island, USA). The pcDNA3 empty vectors were added to all transfections to balance total DNA. The human leukemic cell line MEG-01 was cultured in RPMI1640 medium containing 10% FBS with 100U/ml penicillin and 100ug/ml streptomycin. The generic methyl-transferase inhibitors: Adox (adenosine, periodate oxidized, Sigma, #7154) and MTA (5'-Deoxy-5'-(methylthio) adenosine, Sigma, #D5011) were purchased from Sigma, St. Louis, MO. PMA (20 nM) was added to stimulate Meg-01 cells to differentiate into MKs. The differentiated cells were labeled with anti-CD41 antibody and CD42 antibody (BD) and analyzed by FACS Fortessa machine.

### Coimmunoprecipitation and immunoblot analysis

Transfected 293T cells from 10-cm dish were lysed in 1 ml of lysis buffer (20 mM Hepes, pH 7.9, 150 mM NaCl, 1mM $MgCl_2$, 1% NP40, 10 mM NaF, 0.2 mM $NaVO_4$, 10 mM β-glycerol phosphate) with

freshly added dithiothreitol (DTT; 1 mM) and a proteinase inhibitor cocktail (Roche, Indianapolis, IN). The cells were incubated for 30 min on ice and sonicated with a Bioruptor Sonicator (Diagenode, Denville, NJ). The extracts were cleared by centrifugation at 12,000×g for 15 min at 4°C. Immuno-precipitations were performed at 4°C in lysis buffer in the presence of RNaseA (10μg/ml) using indicated antibody and 50 μl of 50% (V/V) slurry of Protein A agarose (Roche, Indianapolis, IN) or anti-Flag M2 agarose (Sigma, St. Louis, MO) for 4 hr. The precipitates were extensively washed and resuspended in 2×SDS-PAGE sample buffer directly. The immunopurified protein and cell lysates were resolved by 10% SDS-PAGE and transferred to PVDF membranes (Millipore, Billerica, MA) for WB assays. The proteins were visualized by the Immobilon Western Chemiluminescent horseradish peroxidase substrate detection kit (Millipore). PRMT1-V2 specific antibody was reported (*Baldwin et al., 2012*). The other antibodies used were commercially available, including anti-Flag M2 Ab (#F1804, Sigma), RBM15 monoclonal antibody (#66059-1-1g, Proteintech, Chicago), RBM15 polyclonal antibody (#: 10587-1-AP, Proteintech, Chicago), PRMT1 (#07404, Upstate Biotechnology), SF3B1 (#PA5-19679, Thermo Scientific), Ub (#U5397, Sigma) and GAPDH (#MA5-15738, Thermo Scientific, Waltham).

## Detection of in vivo ubiquitylation

HEK293T cells transfected with Flag-RBM15 and His-ubiquitin expression plasmids for 40 hr were treated with 10 μM MG132 for 6 hr before harvesting. Cells were lysed in buffer A (6 M guanidine-HCl, 0.1 M $Na_2HPO_4/NaH_2PO_4$, pH 7.5, 10 mM imidazole) at 4°C for 15 min. The lysate was sonicated for 20 cycles in Bioruptor Sonicator. After cleaning by centrifugation at 12,000g for 30 min at 4°C, the supernatant was incubated with $Ni^{2+}$-NTA beads (QIAGEN, Valencia, CA) for 3 hr at room temperature. The beads were sequentially washed with buffers A, B (1.5 M guanidine-HCl, 25 mM $Na_2HPO_4/NaH_2PO_4$, 20 mM Tris-Cl [pH 6.8], 17.5 mM imidazole), and C (25 mM Tris-Cl [pH 6.8], 20 mM imidazole). The ubiquitin-conjugated proteins were boiled in 2×SDS-PAGE loading buffer containing 200 mM imidazole and subjected to immunoblot analysis.

## Real-time PCR assay

Total RNA was prepared using RNeasy plus Kit (QIAGEN). cDNA was generated by the Verso cDNA synthesis Kit (Thermo Scientific) with random hexamer primers. Real-time PCR assays were performed with Absolute Blue qPCR SYBR green Mix (Thermo Scientific) or Taqman Universal Master Mix II (Applied Biosystems, for GATA1-FL and GATA1-S) on a ViiA 7 system (Applied Biosystems). Primers are listed in *Supplementary file 2*. GAPDH was used as an internal control for normalization. Relative expression level was calculated by Δ(Δct) method and all results were expressed as mean values ± standard errors from at least three independent experiments.

GATA1 isoforms can be detected by regular PCR reactions with the primers: GATA1 Ex1 F: ATCACACTGAGCTTGCCACA, GATA1 Ex3 R: AGCTTGGGAGAGGAATAGGC in *Figure 5H*.

## Identification of protein methylation by mass spectrometry

For identification of methylation sites of targeted proteins, samples were separated by 10% of SDS-PAGE. The gel band were excised from the gel, reduced with 10 mM DTT and alkylated with 55 mM iodoacetamide. Then in-gel digestion was carried out with the sequencing grade modified trypsin (Promega, Fitchburg, WI) in 50 mM ammonium bicarbonate at 37°C overnight. The peptides were extracted twice with 1% trifluoroacetic acid in 50% acetonitrile aqueous solution for 30 min. The extractions were then centrifuged in a speedvac to reduce the volume.

For Liquid chromatography-tandem mass spectrometry (LC-MS/MS) analysis, the digested product was separated by a 60 min gradient elution at a flow rate 0.30 μL/min using an UltiMate 3000 RSLCnano System (Thermo Scientific, USA) which was directly interfaced with a Thermo Q Exactive benchtop mass spectrometer. The analytical column was a home-made fused silica capillary column (75 μm ID, 150 mm length; Upchurch, Oak Harbor, WA) packed with C-18 resin (300 Å, 5 μm, Varian, Lexington, MA). Mobile phase A consisted of 0.1% formic acid and mobile phase B consisted of 100% acetonitrile and 0.1% formic acid. The mass spectrometer was operated in the data-dependent acquisition mode using the Xcalibur 2.2.0 software and there is a single full-scan mass spectrum in the Orbitrap (400–1800 m/z, 70,000 resolution) followed by 8 MS/MS scans under the higher

energy collision dissociation (HCD). The MS/MS spectra from each LC-MS/MS run were searched against the selected database using an in-house Mascot or Proteome Discovery searching algorithm.

## Peptide pulldown assay

The peptide pulldown method is published before (*Vu et al., 2013*). Briefly, two peptides (Biotin-LLYRDRDR$_{ME2A}$DLYPDSDWV and Biotin-LLYRDRDRDLYPDSDWV) with or without methylation on R578 were synthesized. Biotin was added for binding to streptavidin beads. 200 µl of the whole cell extract (10mg/ml in H lysis buffer) prepared as mentioned in co-immunoprecipitation assays was used to incubate with peptides (at final concentration of 1 µM) overnight at 4°C. The beads were washed with wash buffer (300mM NaCl, 1mM DTT, 20mM Tris pH 7.4 and 0.1% NP40) five times. Protein bound to the beads were boiled in SDS sample buffer and resolved in 10% SDS PAGE.

## Survival analysis

RNA-seq data of 200 AML samples were downloaded from the Cancer Genome Atlas (TCGA). The PRMT1 expression levels were used for survival analysis using Kaplan Meier (K-–M) curves and log-rank tests. Samples with available clinical outcomes (survival month, n=103) data were included in the analysis. Logarithm 2-based transformations of each gene were performed prior to any analysis. Significant association was determined at 5% type I error level.

## Target CNOT4 by CRISPR

Two guide RNAs were designed to target exon 2 of human CNOT4 at two different sites using the CRISPR design tool (Zhang's lab) (*Ran et al., 2013b*). The distance between target sites is roughly 100bp. Two gRNAs were cloned into pX330 vector, and transient transfected into 293T cells. Single cells were selected by serial dilution. Single cell colonies were screened for internal deletion by PCR using 2 primers: F: TTCCCCTAAAATGTGTTATGATGA; R: CCAGTGCAGTGTTCTTTCCA. Guide RNA sequences: (1) CACCG GGATGTCATGTCCTCAGCGT; (2) CACCGGTGGATGCCAAAGTG TGCGT.

## Developing anti-peptide antibody against RBM15

RBM15 N-terminal sequence (Acetyl-RTAGRDPVPRRSPRWRRAVPLC) was used to develop rabbit polyclonal antibody against RBM15 by GeneMed Synthesis Inc. (St. Antonio, Texas) for RNA immunoprecipitation assay.

## RNA immunoprecipitation assay

Ten million MEG-01 cells grown at the exponential phase were cross-linked by 1% of formaldehyde in 37°C incubator for 10 min. Cells were washed twice with ice-cold phosphate-buffered saline lysed in H-lysis buffer (20mM HEPES pH 7.9, 300 mM NaCl, 1mM MgCl$_2$, 1% NP40, NaF 10 mM, 0.2 mM NaVO4, 10 mM β-glycerol phosphate, 5% glycerol, 1 mM DTT, 40 U/ml RNaseOUT, and proteinase inhibitors), and then sonicated by Bioruptor (Diagenode) to fragments with average sizes of 200bp (DNA) for 30 min at 4°C. After the lysate was spun down, the supernatant was incubated with RBM15 antibody overnight at 4°C. Beads were washed five times with stringent buffer (50mM Tris pH7.5, 1M NaCl, 1mM ethylenediaminetetraacetic acid [EDTA], 0.1%SDS, 1% Na Deoxycholate, 1M Urea, and 1% NP40), and once with TE buffer (10mM Tris pH 7.9, 1mM EDTA). Immunoprecipitated RNA were eluted by incubating with 150µl RIP elution buffer (50mM Tris-Cl pH 7.9, 5mM EDTA, 1% SDS, 200mM NaCl and 267 µg/ml of proteinase K) for 1h at 45°C. Genomic DNA was removed by DnaseI Turbo (Ambion). RNA was extracted by Trizol (Invitrogen) and cDNA was synthesized with random priming (Thermo Verso cDNA synthesis kit). The cDNA was analyzed by qRT-PCR.

We did two biological replicates for RIP-seq experiments with anti-RBM15 antibody and normal IgG. The RNA was converted into cDNA using SureSelect single strand RNA selection kit (Illumina) and sequenced with Illumina 2000 sequencer.

## Primary cell culture for MK differentiation

Granulocyte-colony stimulating factor mobilized peripheral human blood cells were used to purify CD34$^+$ cells. These CD34$^+$ cells were spinfected with lentiviruses and grown in basic cytokine mix (100ng/ml of stem cell factor (SCF), 100ng/ml FLt3 ligand, 50ng/ml of IL-6 and 20ng/ml of

thrombopoietin[TPO]) with Iscove's Modified Dulbecco's Medium medium plus 20% BIT (Stem Cell Technology, Canada). For MK differentiation, we cultured the CD34$^+$ cells with 50ng/ml TPO and 2ng/ml of SCF. The maturation of MK cells was assessed for the percentage of CD41$^+$CD42$^+$ or CD61$^+$CD42$^+$ populations as well as the percentage of polyploidy with PI staining by FACS analysis.

## Computational analysis of RIP-seq data and RNA-seq data

We first trimmed adaptor sequences and low quality bases from paired end RIP sequencing reads using Trimmomatic (*Bolger et al., 2014*). Processed reads were then mapped to hg19 genome with the STAR (*Dobin et al., 2013*) aligner. RIP reads from all samples were summarized into read coverage profiles on all chromosomes, and peaks were identified with a custom algorithm (*Loeb et al., 2012*) by convolving the read coverage signal with the second derivative of a Gaussian filter (bandwidth = 300). For each peak, we used DESeq (*Anders and Huber, 2010*) to test if there was significant difference in read counts within the peak region between RBM15 samples and control samples after correcting for library size differences. 1515 peaks with more reads in RBM15 samples and FDR < 5% were defined as significant RBM15 binding sites. Each peak was annotated with information including gene name and region (i.e. intron, 3'UTR, 5'UTR, CDS) according to the overlapping RefSeq transcript.

We obtained about 100 million paired-end reads for each RNA sample. All of the reads were mapped to the human reference genome (GRCh37/hg19) using the STAR aligner. A gene annotation file was used to guide the alignments (Ensembl GTF version GRCh37.70). The mean insert sizes and the standard deviations were calculated using Picard-tools (v1.126) (http://broadinstitute.github.io/picard/).

Differential exon usage was calculated by *DEXSeq* (*Anders et al., 2012*) and MISO (*Katz et al., 2010*) programs to measure the abundance of event types: skipped exon (SE), mixed exon usage (MXE), alternative first exon (AFE), alternative last exon (ALE), alternative 5' or 3' splicing sites (A5SS or A3SS) and tandom 3'UTR use (T3UTR). The Read Per Million (RPM) normalized BigWig files were generated using BEDTools (v2.17.0) and bedGraphToBigWig tool (v4). And all the downstream statistical analyses and generating plots were performed in R (v3.1.1) (http://www.r-project.org/). All the RNA-seq and RIP-seq data were deposited in NIH (GSE73893).

## In vitro ubiquitylation assay

In vitro ubiquitination assay was performed essentially as described (*Wang et al., 2004*). Briefly, RBM15 or RBM15 R578K with or without PRMT1-mediated methylation were incubated with HA-CNOT4 in 32.5 μl reaction mixture containing 50 mM Tris (pH7.5), 5 mM MgCl$_2$, 2 mM NaF, 2 mM ATP, 10 mM Okadaic acid, 0.6 M DTT, 0.1 mg E1, 0.2 mg UBCH5C and1 mg recombinant ubiquitin (Sigma, St. Louis, MO). Reactions were carried out for 1 hr at 37°C before stopping with SDS sample loading buffer and resolving by 8% SDS-PAGE. Ubiquitination was detected by Western blot with anti-Flag antibody.

## Acknowledgements

We would like to thank Drs. Nouria Hernandez (UNIL), Stephen Nimer (UMCCC), Michael Crowley, Chris Klug, Kevin Pawlik, Ching-Yi Chen, William Placzek, Camelia Iancu-Rubin (MSH) for help; Dr. John Crispino (NWU), Thomas Mercher (INSERM, Paris) and Dr. Barbara Felber (NIH) for reagents. YL is supported by CNSF (#81270570). We would like to thank the UAB Comprehensive Flow Cytometry Core as part of RDCC (P30AR048311 and P30AI027767) for technical support.

## Additional information

### Funding

| Funder | Grant reference number | Author |
| --- | --- | --- |
| University of Alabama at Birmingham | Start-up fund | Xinyang Zhao |
| National Natural Science Foundation of China | 81270570 | Yanyan Liu |

The funders had no role in study design, data collection and interpretation, or the decision to submit the work for publication.

## Author contributions

LZ, N-TT, HS, ML, XZ, Conception and design, Acquisition of data, Analysis and interpretation of data, Drafting or revising the article; RW, HD, Acquisition of data, Analysis and interpretation of data, Drafting or revising the article; YLu, Analysis and interpretation of data, Acquisition of data, Contributed unpublished essential data or reagents; HT, AG, DZ, KQ, TH, WZ, SL, OAW, RLL, HL, TT, YGZ, Acquisition of data, Analysis and interpretation of data, Contributed unpublished essential data or reagents; SA, Contributed unpublished data and reagents, Acquisition of data, Analysis and interpretation of data, Contributed unpublished essential data or reagents; AKJ, HW, Acquisition of data, Analysis and interpretation of data, Drafting or revising the article, Contributed unpublished essential data or reagents; JC, XH, GR, CL, Analysis and interpretation of data, Drafting or revising the article, Contributed unpublished essential data or reagents; YLi, Acquisition of data, Drafting or revising the article, Contributed unpublished essential data or reagents; DC, Acquisition of data, Analysis and interpretation of data

## Additional files

### Supplementary files

• Supplementary file 1. The real-time PCR primers for human genes.

• Supplementary file 2. shRNA sequences for knocking down CNOT4 and RBM15 genes in human cells.

### Major datasets

The following datasets were generated:

| Author(s) | Year | Dataset title | Dataset URL | Database, license, and accessibility information |
| --- | --- | --- | --- | --- |
| Yuheng Lu, Alireza Khodadadi-Jamayr-an, Xinyang Zhao, Christina Leslie, Tran N | 2015 | Cross-talk between PRMT1-mediated methylation and ubiquitylation on RBM15 controls RNA splicing | http://www.ncbi.nlm.nih.gov/geo/query/acc.cgi?acc=GSE73893 | Publicly available at the NCBI Gene Expression Omnibus (Accession no: GSE73893). |

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
