## [Decision Letter]

Thank you for submitting your work entitled "Cross-talk between PRMT1-mediated arginine methylation and ubiquitylation on RBM15 controls RNA splicing" for peer review at *eLife*. Your submission has been favorably evaluated by Jim Kadonaga (Senior editor) and three reviewers, one of whom, Michael Green, is a member of our Board of Reviewing Editors.

The reviewers have discussed the reviews with one another and the Reviewing editor has drafted this decision to help you prepare a revised submission.

Summary:

In this manuscript Zhang et al. show that the RNA binding protein RBM15 is methylated by PRMT1 resulting in RBM15 degradation. They show that RMB15 binds to pre-mRNAs that are important for megakaryocytic development. They present evidence indicating that RBM15 recruits the general splicing factor SF3B1. They show that during megakaryocytic development PRMT1 is down-regulated leading to up-regulation of RBM15.

The involvement of PRMT1 and RBM15 in cell differentiation via regulation of alternative splicing are exciting findings, the less satisfying parts of the manuscript include: i) the rather limited analysis of the mechanisms by which the RBM15/PRMT1 axis regulates RNA metabolism and ii) the absence of clear evidence that the changes in RNA processing associated with RBM15 activity are indeed necessary/sufficient to contribute to megakaryocyte differentiation.

Essential revisions:

1) The role of PRMT1 in regulating RBM15 levels is very convincing. The conclusion that RBM15 functions by recruiting SF3B1 is less convincing.

Of multiple general splicing factors found to interact with RBM15, what is the rationale for focusing on RBM15 and not for example U2AF?

Is the loss of SF3B1 binding following knockdown of RBM15 specific to SF3B1? If other general splicing factors were analyzed similarly would the results be the same or different?

Were the coimmunoprecipitations used to detect the RBM15-SF3B1 interaction performed in the presence of RNAse used to rule out RNA bridging effects?

2) The authors propose that RBM15 affects megakaryocytic development by regulating splicing of specific pre-mRNAs such as MPL1 and GATA1. As a control, can the authors show that knockdown of MPL1 and GATA1 leads to a similar cellular phenotype as knockdown of RBM15.

3) Figure 5 reveals enrichment of RBM9/RBFOX2 and PABPC4 motifs among RBM15 RIP-Seq targets. PABPC4 is shown to interact with RBM15 ( [Supplementary-material SD4-data]) suggesting a potential explanation for the recruitment of RBM15 to PABPC4-bound 3' UTRs. But the relationship with RBFOX2 remains unexplained. In this regard it would be important to analyze the overlap between transcriptome-wide (by RNA-Seq) changes induced by RBM15 knock down and those induced by RBFOX2 or PABPC4. Also interesting would be overlap between RBM15 targets determined by RIP-Seq and by RNA-Seq upon RBM15 knock down.

4) While the results of Figure 6 are consistent with the possibility that RBM15/PRMT1 control differentiation by modulating alternative splicing of target genes like GATA1, RUNX1 and/or c-MPL, these data do not formally demonstrate that this is the case. It is clear that knock down or overexpression of RBM15 can regulate alternative splicing of these genes, but are the actual changes in RBM15 protein levels expression observed during differentiation sufficient to explain the observed changes in splicing? And are the observed changes in splicing sufficient to modulate aspects of cell differentiation? While it is possible that RBM15 coordinates multiple events, each contributing to differentiation, previous work in cancer cells has shown that modulation of specific, prominent alternative splicing events by 2'-O-methyl-phosphothiate antisense oligonucleotides can influence cellular phenotypes in a background of multiple transcriptome changes (Bechara Mol Cell 2013; Maimon et al Cell Reports 2014).

5) In Figure 1, the overall quality of dimethyl-R WB is not good enough, especially in Figure 1. The authors should obtain stronger WB signals, if possible.

6) In Figure 3, "DB75" label should be replaced with "MG132" which is then consistent with the text in the manuscript.

[Editors' note: further revisions were requested prior to acceptance, as described below.]

Thank you for resubmitting your work entitled "Cross-talk between PRMT1-mediated arginine methylation and ubiquitylation on RBM15 controls RNA splicing" for further consideration at *eLife*. Your revised article has been favorably evaluated by Randy Schekman (Senior editor), Michael Green (Reviewing editor) and the two originial expert reviewers. The reviewers agree that the manuscript has been improved but there are a few remaining issues that need to be addressed before acceptance, as outlined below:

1) While the authors do not demonstrate that alternative splicing changes in genes relevant for megakaryocyte differentiation are behind the effects of RBM15 in this process, the multiple bona fide megakaryocyte differentiation factors affected and the magnitude of some of these changes make their hypothesis quite reasonable. We are a bit surprised, however, that, given the results of RUNX knock down provided in the revision, the authors did not attempt to rescue the effects of RBM15 knock down by overexpression of RUNX1.

2) Data demonstrating the specific effects of RBM15 on SF3B1 recruitment are not provided. The authors should at least acknowledge – as they did in their rebuttal – that their results of SF3B1 recruitment may just serve as a surrogate measure of recruitment of 3' splice site-recognizing factors.

3) The extent of the overlap between RIP-Seq and RNA-Seq data, particularly regarding megakaryocyte differentiation-related genes could be highlighted.

---

## [Author Response]

*[…] The involvement of PRMT1 and RBM15 in cell differentiation via regulation of alternative splicing are exciting findings, the less satisfying parts of the manuscript include: i) the rather limited analysis of the mechanisms by which the RBM15/PRMT1 axis regulates RNA metabolism and ii) the absence of clear evidence that the changes in RNA processing associated with RBM15 activity are indeed necessary/sufficient to contribute to megakaryocyte differentiation.*

We have now included RNA-seq data by comparing the exon usage changes upon RBM15 knock-down in order to obtain an unbiased assessment of splicing changes dependent on RBM15. Given that RBM15-mediated splicing regulation controls the three key transcription factors RUNX1, TAL1 and GATA1 for megakaryocyte (MK) differentiation, our data provides compelling evidence that RBM15-mediated splicing regulates differentiation. We agree that additional experiments would be required to determine the precise contribution of RBM15-dependent events relative to other drivers of MK differentiation. Similarly, to identify precisely which RBM15-regulated alternative splicing event(s) is specifically implicated in MK differentiation is an important question, but not a trivial one, and we feel that addressing this would be beyond the scope of the current study. We have toned down our statements and conclusions regarding this aspect of our results. Our data support that RBM15 contributes to but is not solely sufficient for MK development.

*Essential revisions: 1) The role of PRMT1 in regulating RBM15 levels is very convincing. The conclusion that RBM15 functions by recruiting SF3B1 is less convincing. Of multiple general splicing factors found to interact with RBM15, what is the rationale for focusing on RBM15 and not for example U2AF? Is the loss of SF3B1 binding following knockdown of RBM15 specific to SF3B1? If other general splicing factors were analyzed similarly would the results be the same or different? Were the coimmunoprecipitations used to detect the RBM15-SF3B1 interaction performed in the presence of RNAse used to rule out RNA bridging effects?*

We would like to thank the reviewers for pointing out the role of PRMT1 in regulating RBM15 levels very convincing. The immunofluorescent staining data from Horiuchi et al., 2013, as well as from our unpublished immunofluorescent staining data demonstrated that RBM15 colocalizes with splicing speckles, which implies that RBM15 is involved in pre-mRNA splicing or at least in some step along the mRNA maturation pathway. By clustering proteins found in mass spectrometry analysis, we showed that a large proportion of RBM15-associated proteins are involved in many steps of RNA processing. Given that RBM15 binds to intronic regions, we focused on SF3B1 which is a core general splicing factor interacting with introns. In addition, SF3B1 is often mutated in myeloid dysplasia syndrome a pre-leukemic disease often associated with defects in MK development. Moreover, since SF3B1 is a component of the U2 snRNP, which itself depends on previous recruitment of U2AF to the polypyrimidine stretch/3’AG region in order to stably interact with the branch point, we reasoned that assessing SF3B1 binding should serve as a surrogate measure of constitutive splicing factor interactions with 3’ splice sites. That being said, U2AF is mutated in hematological malignancies as well. Intriguingly mutations in U2AF and SF3B1 are often mutually exclusive from each other in hematological malignancies (Patnaik et al., 2013). Both U2AFand SF3B1 are within the A complex, but RBM15 might bind to U2AF in the E complex before interacting with the A complex. Further biochemistry experiments with a splicing minigene reporter and in vitro splicing assays would be required to dissect the precise molecular mechanism by which RBM15 regulates splicing, which we are definitely planning to pursue with our collaborators, but which we feel, falls beyond the scope of the current manuscript.

We would like to point out that we did mention in the Methods section (subheading “Coimmunoprecipitation and Immunoblot analysis”) that we included RNase A for the co-immunoprecipitation assays. So the interaction between SF3B1 and RBM15 is a protein-protein interaction, not mediated via an RNA moiety, although we cannot exclude at this point the possibility that the interaction might be mediated via other protein subunits in the U2 snRNP.

2) The authors propose that RBM15 affects megakaryocytic development by regulating splicing of specific pre-mRNAs such as MPL1 and GATA1. As a control, can the authors show that knockdown of MPL1 and GATA1 leads to a similar cellular phenotype as knockdown of RBM15.

We did not knockdown MPL and GATA1 in our assays. However we cite compelling evidences from literature including using mouse knockout models showing that *RUNX1, MPL, TAL1* and *GATA1* are essential genes for megakaryocyte differentiation (Lacombe et al., 2013; Pimkin et al., 2014). We knocked down RUNX1 in CD34^+^ cells and we did observe the defects in megakaryocyte differentiation (Figure 8). Given that *RBM15* controls the alternative splicing of these essential genes for megakaryocyte differentiation including *TAL1* as we have now shown by RNA-seq analysis in the revised manuscript, it would be very challenging to correct one gene’s alternative splicing to restore the megakaryocyte differentiation defects caused by RBM15 knockdown. It was previously reported (Xiao et. al. 2014) by our collaborator Dr. Raffel that overexpression of MPL alone fails to rescue the MK differentiation of RBM15 knock-out cells, so we envisage that it might be the joint contribution of more than one RBM15 target that regulates this process.

Author response image 1.Knockdown RUNX1 in CD34^+^ cord blood cells block megakaryocyte differentiation.Two shRNA against RUNX1 were expressed by retroviruses. The cells were grown in pro-megakaryocyte differentiation medium and harvested on day five.**DOI:**
http://dx.doi.org/10.7554/eLife.07938.031

*3) Figure 5 reveals enrichment of RBM9/RBFOX2 and PABPC4 motifs among RBM15 RIP-Seq targets. PABPC4 is shown to interact with RBM15 ([Supplementary-material SD4-data]), suggesting a potential explanation for the recruitment of RBM15 to PABPC4-bound 3' UTRs. But the relationship with RBFOX2 remains unexplained. In this regard it would be important to analyze the overlap between transcriptome-wide (by RNA-Seq) changes induced by RBM15 knock down and those induced by RBFOX2 or PABPC4. Also interesting would be overlap between RBM15 targets determined by RIP-Seq and by RNA-Seq upon RBM15 knock down.*

The roles of RBFOX2 and PABPC4 in hematopoiesis are not known. The interactions of RBFOX2 or PABPC4 with the RBM15-bound RNA elements are pure predictions of bioinformatics analysis. We have shown that PABPC4 interacts with RBM15 in mass spectrometry analysis. Whether RBFOX2 interacts with RBM15 protein is not known. We will need to validate the interactions with co-immunoprecipitation assays and gene knockdown assays. So in the revised manuscript, we deleted the predicted motifs for RBFOX2 and PABPC4. We agree with the reviewers that finding the overlapping targets determined by RIP-seq and by RNA-seq upon RBM15 knockdown is very interesting and needed to understand the roles of RBM15 in RNA metabolism. We have now included this analysis in the revised manuscript, leaving out the more speculative data and discussion regarding RBFOX2 and PABPC4.

*4) While the results of Figure 6 are consistent with the possibility that RBM15/PRMT1 control differentiation by modulating alternative splicing of target genes like GATA1, RUNX1 and/or c-MPL, these data do not formally demonstrate that this is the case. It is clear that knock down or overexpression of RBM15 can regulate alternative splicing of these genes, but are the actual changes in RBM15 protein levels expression observed during differentiation sufficient to explain the observed changes in splicing? And are the observed changes in splicing sufficient to modulate aspects of cell differentiation? While it is possible that RBM15 coordinates multiple events, each contributing to differentiation, previous work in cancer cells has shown that modulation of specific, prominent alternative splicing events by 2'-O-methyl-phosphothiate antisense oligonucleotides can influence cellular phenotypes in a background of multiple transcriptome changes (Bechara Mol Cell 2013; Maimon et al Cell Reports 2014).*

In Figure 6, we showed that during megakaryocyte differentiation of primary CD34^+^ cells alternative splicing changes in the same fashion as directly manipulating RBM15 dosages in leukemia cells. When we treated the primary cells with PRMT1 inhibitors, we observed that the splicing patterns change like that in RBM15 overexpression.

Our data showed that multiple splicing events regulated by RBM15 collectively stimulate MK differentiation. It is hard to conclude that splicing regulation is sufficient for MK differentiation, since RBM15 also participates in regulations of RNA export, and as well in epigenetic regulation such as binding to Xist RNA, controlling histone H3K4 trimethylation (our unpublished data). One of our co-author Dr. Glen Raffel has tried and failed to rescue MK differentiation using the RBM15 knockout cells by overexpressing MPL gene. The results were presented in the blood paper (Xiao et al., 2014). Given that multiple transcription factors genes essential for megakaryocyte differentiation such as *GATA1, RUNX1, TAL1, GFIB* and *FLI1* are affected and knockout of any one of these transcription factors will lead to MK differentiation defect, correcting splicing of one transcription factor may not be enough. Using oligonucleotides to correct RNA splicing defects has been used successfully for muscle diseases, and would be a very interesting and an important experiment to do with therapeutic potentials in acute megakaryocytic leukemia with chromosome translocation t(1;22) which produces RBM15-MKl1 fusion. To give definitive answer to this question needs to do more screening to narrow down which gene can rescue MK differentiation by cDNA overexpression first before designing oligos to correct splicing.

*5) In Figure 1, the overall quality of dimethyl-R WB is not good enough, especially in Figure 1, G. The authors should obtain stronger WB signals, if possible.*

We have now included a new set of data (new Figure 1—figure supplement 4) with a custom antibody specifically recognizing dimethylated RBM15.

*6) In Figure 3, "DB75" label should be replaced with "MG132" which is then consistent with the text in the manuscript.*

Thank you for pointing this out. We have made the correction.

References:

Lacombe, J., Krosl, G., Tremblay, M., Gerby, B., Martin, R., Aplan, P.D., Lemieux, S., and Hoang, T. (2013). Genetic interaction between Kit and Scl. Blood 122, 1150-1161.

Patnaik, M.M., Lasho, T.L., Finke, C.M., Hanson, C.A., Hodnefield, J.M., Knudson, R.A., Ketterling, R.P., Pardanani, A., and Tefferi, A. (2013). Spliceosome mutations involving SRSF2, SF3B1, and U2AF35 in chronic myelomonocytic leukemia: prevalence, clinical correlates, and prognostic relevance. Am J Hematol 88, 201-206.

Pimkin, M., Kossenkov, A.V., Mishra, T., Morrissey, C.S., Wu, W., Keller, C.A., Blobel, G.A., Lee, D., Beer, M.A., Hardison, R.C., et al. (2014). Divergent functions of hematopoietic transcription factors in lineage priming and differentiation during erythro-megakaryopoiesis. Genome Res 24, 1932-1944.

[Editors' note: further revisions were requested prior to acceptance, as described below.]

*1) While the authors do not demonstrate that alternative splicing changes in genes relevant for megakaryocyte differentiation are behind the effects of RBM15 in this process, the multiple bona fide megakaryocyte differentiation factors affected and the magnitude of some of these changes make their hypothesis quite reasonable. We are a bit surprised, however, that, given the results of RUNX knock down provided in the revision, the authors did not attempt to rescue the effects of RBM15 knock down by overexpression of RUNX1.*

A set of key transcription factors including RUNX1, TAL1 and GATA1 interact with each other in a protein complex to regulate megakaryocyte (MK) differentiation (Pimkin et al., 2014; Tijssen et al., 2011). So knocking down any one of these transcription factors such as RUNX1 blocks megakaryocyte differentiation as we showed to the reviewers. On the other hand, overexpressing any of these transcription factors individually may not rescue MK differentiation due to stoichiometric changes of transcription factors binding to a given promoter.

Dr. Lausen’s group has demonstrated that forced expression of RUNX1 did promote MK differentiation (Kuvardina et al., 2015). From our own data, we know that RUNX1 directly activates the transcription of RBM15 gene. Here in panel A, we knocked down RUNX1 by two different shRNAs in HEL cells. We detected the expression levels of RUNX1 and RBM15 were decreased in the RUNX1 knockdown cells by real time PCR assays. Furthermore, we detected that RUNX1 binds to RBM15 promoter specifically in chromatin immunoprecipitation assays (ChIP), as controls RUNX1 did not bind to RBM15 promoter in HEL cells with RUNX1 knock down. These results were consistent with RUNX1 and RUNX1-ETO ChIP-seq data in literature (GSE24674, GSE42075, GSE46044, GSE43834). Thus, RUNX1-mediated rescue of MK differentiation in RBM15 knockdown cells is not due to the correction of RBM15-mediated RNA splicing but rather due to the activation of RBM15 gene expression by RUNX1. Increased expression of RBM15 by RUNX1-mediated transcriptional activation will further correct the alternative splicing of genes like GATA1 and TAL1 etc.

Author response image 2.**DOI:**
http://dx.doi.org/10.7554/eLife.07938.032

*2) Data demonstrating the specific effects of RBM15 on SF3B1 recruitment are not provided. The authors should at least acknowledge – as they did in their rebuttal – that their results of SF3B1 recruitment may just serve as a surrogate measure of recruitment of 3' splice site-recognizing factors*.

We have changed the text in the manuscript as suggested in the subection “RBM15 controls alternative splicing via its interaction with SF3B1”.

*3) The extent of the overlap between RIP-Seq and RNA-Seq data, particularly regarding megakaryocyte differentiation-related genes could be highlighted.*

We have highlighted the genes related to MK differentiation in excel sheets ([Supplementary-material SD1-data], [Supplementary-material SD2-data], [Supplementary-material SD3-data]), and changed the text in the manuscript accordingly.

Reference:

Kuvardina, O.N., Herglotz, J., Kolodziej, S., Kohrs, N., Herkt, S., Wojcik, B., Oellerich, T., Corso, J., Behrens, K., Kumar, A., et al. (2015). RUNX1 represses the erythroid gene expression program during megakaryocytic differentiation. Blood 125, 3570-3579.